# Didemnin B and ternatin-4 differentially inhibit conformational changes in eEF1A required for aminoacyl-tRNA accommodation into mammalian ribosomes

**Manuel F Juette**[1†], **Jordan D Carelli**[2†], **Emily J Rundlet**[1,3,4†], **Alan Brown**[5‡], **Sichen Shao**[5§], **Angelica Ferguson**[1], **Michael R Wasserman**[1], **Mikael Holm**[1,3], **Jack Taunton**[2]*, **Scott C Blanchard**[1,3]*

[1]Department of Physiology and Biophysics, Weill Cornell Medicine, New York, United States; [2]Chemistry and Chemical Biology Graduate Program, University of California, San Francisco, San Francisco, United States; [3]Department of Structural Biology, St. Jude Children's Research Hospital, Memphis, United States; [4]Tri-Institutional PhD Program in Chemical Biology, Weill Cornell Medicine, New York, United States; [5]MRC-LMB, Francis Crick Avenue, Cambridge, United Kingdom

*For correspondence:
jack.taunton@ucsf.edu (JT);
Scott.Blanchard@stjude.org
(SCB)

†These authors contributed
equally to this work

**Present address:** ‡Department
of Biological Chemistry and
Molecular Pharmacology,
Blavatnik Institute, Harvard
Medical School, Boston, United
States; §Department of Cell
Biology, Blavatnik Institute,
Harvard Medical School, Boston,
United States

**Competing interest:** See page
23

**Reviewing Editor:** Alan G
Hinnebusch, Eunice Kennedy
Shriver National Institute of Child
Health and Human Development,
United States

**Abstract** Rapid and accurate mRNA translation requires efficient codon-dependent delivery of the correct aminoacyl-tRNA (aa-tRNA) to the ribosomal A site. In mammals, this fidelity-determining reaction is facilitated by the GTPase elongation factor-1 alpha (eEF1A), which escorts aa-tRNA as an eEF1A(GTP)-aa-tRNA ternary complex into the ribosome. The structurally unrelated cyclic peptides didemnin B and ternatin-4 bind to the eEF1A(GTP)-aa-tRNA ternary complex and inhibit translation but have different effects on protein synthesis in vitro and in vivo. Here, we employ single-molecule fluorescence imaging and cryogenic electron microscopy to determine how these natural products inhibit translational elongation on mammalian ribosomes. By binding to a common site on eEF1A, didemnin B and ternatin-4 trap eEF1A in an intermediate state of aa-tRNA selection, preventing eEF1A release and aa-tRNA accommodation on the ribosome. We also show that didemnin B and ternatin-4 exhibit distinct effects on the dynamics of aa-tRNA selection that inform on observed disparities in their inhibition efficacies and physiological impacts. These integrated findings underscore the value of dynamics measurements in assessing the mechanism of small-molecule inhibition and highlight potential of single-molecule methods to reveal how distinct natural products differentially impact the human translation mechanism.

## Editor's evaluation

Juette and coworkers employed single-molecule fluorescence, cryogenic-electron microscopy structures, and in vivo measurements to investigate the mechanism whereby two natural products with potential as cancer therapeutics (didemnin B and ternatin-4) inhibit accommodation of tRNA within the ribosomal A site during translation elongation. Their results demonstrate convincingly that both molecules inhibit tRNA accommodation by interfering with the movement of eukaryotic elongation factor 1 α after its activation by the GTPase activation site of the ribosome, but that the degree and nature of this restriction are subtly different between the two, leading to more marked differences in their effects on global translation and cell growth. The results of this important and

valuable interdisciplinary work solidify prior conclusions, particularly on didemnin B, and illuminate the similarities and differences in how these two drugs interfere with the normal functioning of the elongating ribosome in vitro and inhibit protein synthesis and cell growth in vivo.

## Introduction

Translation of the genetic code from mRNA into protein is a multi-step process catalyzed by the two-subunit ribosome (80 S in eukaryotes) in coordination with translational GTPases (*Behrmann et al., 2015*). Each translation step is regulated by signaling pathways linked to cell growth, differentiation, nutrient sensing, and homeostatic quality control. Protein synthesis status is thus a central hub for sensing cellular stress. Dysregulated protein synthesis plays a role in several human diseases and is a therapeutic vulnerability in cancer and viral infection (*Bhat et al., 2015*; *Hoang et al., 2021*; *Xu and Ruggero, 2020*).

The elongation cycle in eukaryotic protein synthesis begins with the binding of a ternary complex of the highly conserved, three-domain (DI-III) eukaryotic elongation factor-1 alpha (eEF1A), GTP, and aminoacyl-tRNA (aa-tRNA) to the Aminoacyl (A) site at the leading edge of the eukaryotic ribosome (80 S; *Figure 1A*; *Abbas et al., 2015*). Within the small subunit, base-pairing interactions between the A-site mRNA codon and a cognate aa-tRNA anticodon trigger a sequence of structural rearrangements that dock eEF1A at the large subunit GTPase activating center (*Voorhees et al., 2010*). There, the large subunit GTPase activating center triggers eEF1A to hydrolyze GTP, ultimately driving eEF1A dissociation and accommodation of the aa-tRNA 3'-CCA end into the large subunit peptidyl transferase center (*Budkevich et al., 2014*; *Ferguson et al., 2015*; *Voorhees et al., 2010*). Once fully accommodated, aa-tRNA undergoes a peptide bond-forming condensation reaction that extends the nascent polypeptide by one amino acid, generating a 'classical' pre-translocation ribosome complex. Ensuing conformational processes within the ribosome, including intersubunit rotation and tRNA translocation with respect to the large subunit ("hybrid"-state formation), enable engagement by eukaryotic elongation factor-2 (eEF2), which catalyzes mRNA and tRNA translocation with respect to the small subunit to complete the elongation cycle (*Noller et al., 2017*). Processive elongation reactions often repeat over hundreds to thousands of mRNA codons to synthesize a single protein, making elongation reactions particularly sensitive to modest structural or kinetic perturbations.

Investigations of aa-tRNA selection with human and bacterial ribosomes have revealed conformational sampling of aa-tRNA between distinct positions within the A site of the ribosome en route to peptide bond formation: codon-recognition (CR), GTPase-activated (GA), and fully accommodated (AC) states (*Figure 1A*; *Blanchard et al., 2004a*; *Ferguson et al., 2015*; *Geggier et al., 2010*; *Ieong et al., 2016*; *Morse et al., 2020*; *Whitford et al., 2010*). Work in bacteria established a two-step kinetic proofreading mechanism of decoding fidelity demarcated by GTP hydrolysis (*Geggier et al., 2010*; *Hopfield, 1974*; *Morse et al., 2020*; *Ninio, 1975*; *Pape et al., 1998*; *Whitford et al., 2010*). In this model, relatively rapid, reversible transitions between CR and GA-like states precede GTP hydrolysis during the 'initial selection' step, while relatively slow, rate-limiting reversible transitions between GA- and AC-like states follow GTP hydrolysis during the 'proofreading' step (*Figure 1A*; *Geggier et al., 2010*; *Morse et al., 2020*; *Whitford et al., 2010*). Similar concepts were put forward through structural studies of the bacterial antibiotic kirromycin, which targets the bacterial homolog of eEF1A (EF-Tu) (*Fischer et al., 2015*; *Schmeing et al., 2009*).

Analogous to kirromycin, multiple natural products identified in phenotypic screens for anticancer or other biological activities directly target human eEF1A, suggesting a therapeutic potential of translation elongation inhibition in combating human malignancy (*Burgers and Fürst, 2021*; *Carelli et al., 2015*; *Crews et al., 1994*; *Klein et al., 2021*; *Krastel et al., 2015*; *Lindqvist et al., 2010*; *Sun et al., 2021*). Of these, didemnin B (henceforth 'didemnin') and its variants have been studied most extensively (*Hossain et al., 1988*; *Li et al., 1984*; *Rinehart et al., 1981*), including clinical trials for the treatment of specific cancer indications (*Kucuk et al., 2000*) and severe acute respiratory syndrome coronavirus 2 (SARS-CoV-2 or COVID-19) (*White et al., 2021*). Ternatins, a family of cyclic peptides chemically unrelated to didemnin (*Figure 1—figure supplement 1*), bind eEF1A competitively with didemnin and inhibit translation elongation (*Carelli et al., 2015*; *Ito et al., 2009*; *Kobayashi et al., 2012*; *Shimokawa et al., 2008*). Ternatin-4, a synthetic ternatin variant with 500-fold increased cytotoxic potency over the parent compound (*Carelli et al., 2015*), has also been shown to reduce viral

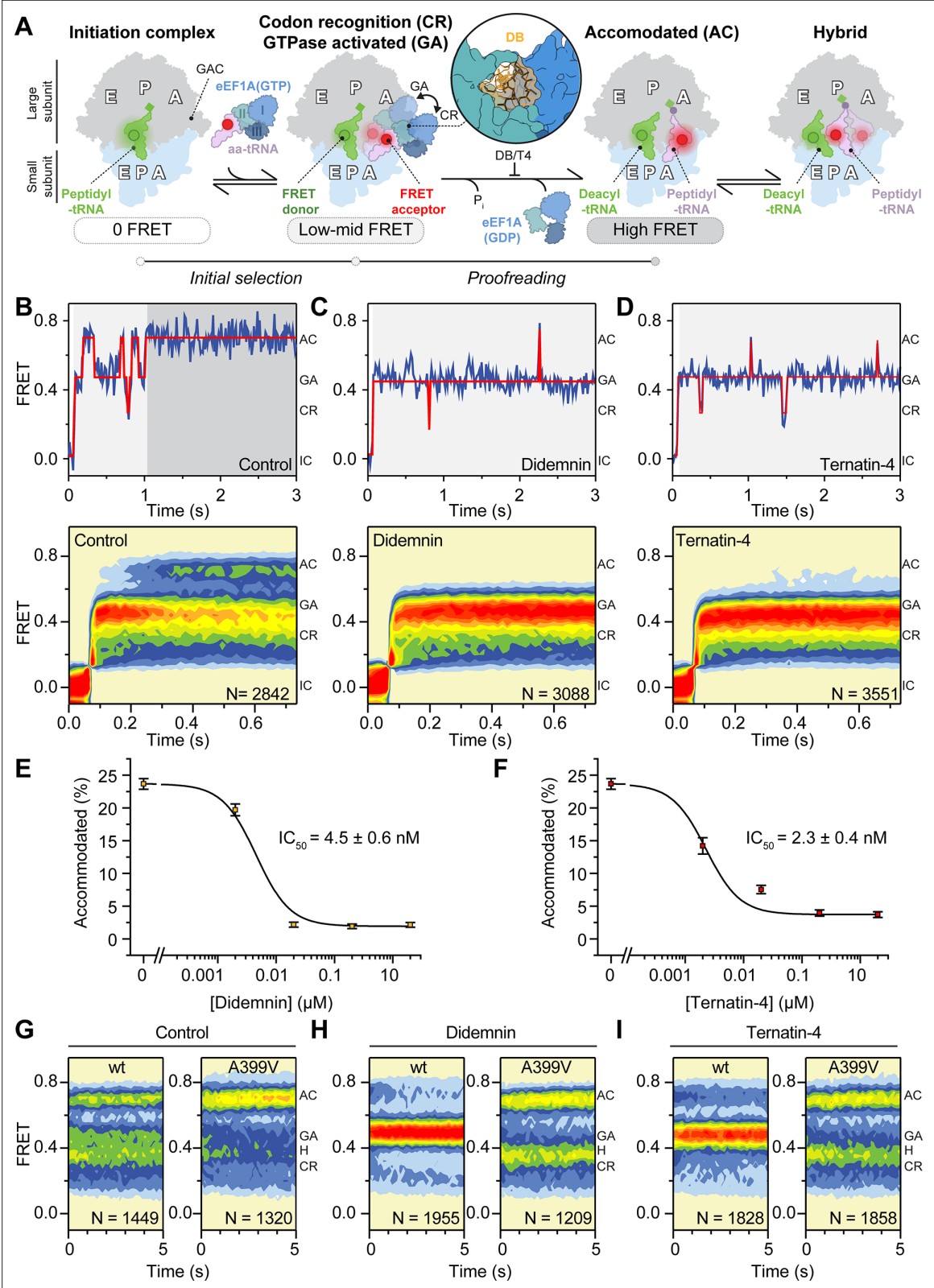

**Figure 1.** Mechanism of didemnin and ternatin-4 inhibition revealed by smFRET. (**A**) Schematic of the experimental setup. Acceptor Cy5 (red circle)-labeled eEF1A(GTP)-aa-tRNA ternary complex is delivered to 80 S initiation complexes with donor Cy3 (green circle)-labeled P-site tRNA. eEF1A is colored by domain (I-III). Codon recognition (CR, low FRET,~0.2, solid) leads to a pre-accommodated GTPase-activated (GA, mid FRET,~0.45, transparent) state, where eEF1A docks on the large subunit GTPase activating center (GAC). Accommodation (AC) and peptide bond formation

*Figure 1 continued on next page*

*Figure 1 continued*

produce a pre-translocation complex, which samples classical (high-FRET, ~0.7) and hybrid (H, mid-FRET, ~0.25–0.4) conformations in equilibrium. Inset shows the didemnin (DB, orange) and ternatin-4 (**T4**) binding site on eEF1A (PDB-ID: 5LZS) (*Shao et al., 2016*). (**B–D**) Representative smFRET traces, (*top*) and post-synchronized population histograms of N traces (*bottom*) of accommodation dynamics of pre-steady state reactions in the presence of (**B**) DMSO (control) or 20 µM (**C**) didemnin or (**D**) ternatin-4. Dotted circles and shading behind traces indicate the state assignment as described in (A; white, unbound; light gray, CR/GA; dark gray, accommodated). (**E, F**) Dose-response curves of the accommodated fraction in the presence of (**E**) didemnin or (**F**) ternatin-4. Error bars: s.e.m. from 1000 bootstrap samples. (**G–I**) Population histograms of N traces for steady-state reactions, formed with ternary complex containing recombinant wild-type (wt) or A399V eEF1A. See also *Figure 1—figure supplements 1–3*.

The online version of this article includes the following source data and figure supplement(s) for figure 1:

**Figure supplement 1.** Molecular structures of (**A**) didemnin (didemnin B) and (**B**) ternatin-4.

**Figure supplement 2.** Dose response profiles of the accommodated fraction for (**A**) intermediately active ternatin-3, and (**B**) inactive ternatin-2.

**Figure supplement 3.** Recombinant expression of Flag-tagged eEF1A.

**Figure supplement 3—source data 1.** Raw image file of the coomassie-stained SDS-PAGE gel for *Figure 1—figure supplement 3A*.

infectivity of SARS-CoV-2 (*Gordon et al., 2020*). Of the divergent natural product inhibitors of eEF1A, ternatins represent an attractive class of lead compounds due to their accessibility by total synthesis (*Carelli et al., 2015*).

Didemnin binds between DI and DIII to stall eEF1A(GDP) on the ribosome during aa-tRNA selection, as revealed by a cryogenic electron microscopy (cryo-EM) reconstruction of elongating rabbit reticulocyte lysate ribosomes (*Figure 1A*; *Shao et al., 2016*). Didemnin traps eEF1A in an active, GTP-bound conformation even after GTP hydrolysis and $P_i$ release, acting comparably to kirromycin on EF-Tu (*Fischer et al., 2015*; *Schmeing et al., 2009*). Although ternatin-4 competitively inhibits didemnin binding (*Carelli et al., 2015*), ternatin-4's binding site has not been determined structurally, and comparative mechanistic studies of didemnin and ternatins have yet to be performed.

Here, we use single-molecule fluorescence resonance energy transfer (smFRET) imaging and comparative cryo-EM structural analysis of partially and fully reconstituted mammalian ribosome complexes to elucidate the effects of didemnin and ternatin-4 on aa-tRNA selection. This smFRET platform provided quantitative structural and kinetic data that further define the aa-tRNA selection reaction coordinate on human ribosomes under uninhibited and inhibited conditions. These include the rates of eEF1A(GTP)-aa-tRNA ternary complex binding and the stepwise progression of aa-tRNA through distinct initial selection (CR-to-GA) and proofreading (GA-to-AC) stages within the A site of the ribosome en route to peptide bond formation (*Blanchard et al., 2004a*; *Ferguson et al., 2015*; *Geggier et al., 2010*; *Juette et al., 2016*). We show that, despite sharing the same allosteric binding site on eEF1A, didemnin and ternatin-4 differentially perturb the conformational dynamics of ribosome-associated eEF1A(GTP)-aa-tRNA ternary complex in ways that correlate with their effects on cellular growth and protein synthesis. These observations shed light on how these drugs impact the rate-determining conformational changes in eEF1A governing aa-tRNA accommodation prior to peptide bond formation and reveal that their distinct pharmacological effects likely derive from small differences in their binding kinetics and eEF1A-bound dynamics.

## Results

### Didemnin and ternatin-4 inhibit aa-tRNA accommodation

We set out to define and compare the impacts of didemnin and ternatin-4 on the mechanism of aa-tRNA selection on human ribosomes using a smFRET platform that monitors the change in distance between fluorescently labeled incoming aa-tRNA and P-site tRNA (*Blanchard et al., 2004a*; *Ferguson et al., 2015*; *Geggier et al., 2010*; *Juette et al., 2016*). This smFRET platform enables interrogation of dynamics within purified, reconstituted human translation elongation reactions (*Ferguson et al., 2015*). Functional human 80 S initiation complexes were reconstituted from ribosomal subunits isolated from HEK293T cells, synthetic mRNA, and fluorescently labeled initiator tRNA in the P-site (*Ferguson et al., 2015*; Materials and methods). Human initiation complexes were surface-tethered within passivated microfluidic flow cells by a biotin-streptavidin bridge on the mRNA 5'-end (*Juette et al., 2016*). Ternary complex was formed with eEF1A purified from rabbit reticulocyte lysate (identical in primary sequence to human eEF1A1), fluorescently labeled Phe-tRNA[Phe], and GTP (Materials

and methods). Pre-formed eEF1A(GTP)Phe-tRNA$^{Phe}$ ternary complex was stopped-flow delivered to the immobilized initiation complexes while imaging in real time to assess the specific effects of didemnin and ternatin-4 on the progression through CR, GA, and AC states during aa-tRNA selection (Materials and methods).

In the absence of inhibitor, we observed a stepwise progression of aa-tRNA into the A site (*Figure 1A and B*). As has been described previously, the ultimate high (~0.7) FRET accommodated state was consistent with a classical position of the Met-Phe-tRNA$^{Phe}$ within the small and large subunit A sites ('A/A' configuration) of the pre-translocation complex after peptide bond formation (*Figure 1A and B*; *Ferguson et al., 2015*; *Geggier et al., 2010*; *Munro et al., 2007*; *Rundlet et al., 2021*; *Whitford et al., 2010*). Also in line with prior investigations in bacteria (*Blanchard et al., 2004b*; *Geggier et al., 2010*; *Morse et al., 2020*) and human (*Ferguson et al., 2015*), entrance into the final AC state was preceded by transient, reversible movements through two key intermediates in the aa-tRNA selection process characterized by low- (CR,~0.2) and intermediate- (GA,~0.45) FRET efficiencies (*Figure 1A and B*).

In the presence of 20 µM didemnin or ternatin-4, ribosome complexes efficiently stalled in a relatively long-lived, GA-like (~0.45 FRET efficiency) state (*Figure 1C and D*). These findings are consistent with the cryo-EM structure of a didemnin-stalled rabbit elongation complex containing a classical peptidyl-tRNA in the P site ('P/P' configuration) and aa-tRNA in a bent 'A/T' configuration bound to eEF1A at the A-site subunit interface (*Shao et al., 2016*). Quantifying the fraction of smFRET trajectories that rapidly achieved the high-FRET accommodated state (thus considered to be molecules that escaped drug inhibition), revealed similar dose-dependent inhibition profiles for didemnin and ternatin-4, with IC$_{50}$ values of 4.5±0.6 nM and 2.3±0.4 nM, respectively (*Figure 1E and F*). Two different ternatin variants showed complete inactivity (ternatin-2) and ~5-fold less potency than ternatin-4 (ternatin-3, *Figure 1—figure supplement 2*), consistent with both eEF1A binding and cell proliferation assays (*Carelli et al., 2015*). These data argue that didemnin and ternatin trap eEF1A during intermediate states of aa-tRNA selection by slowing processes required for aa-tRNA accommodation from a timescale of hundreds of milliseconds to minutes.

The A399V mutation in eEF1A DIII adjacent to the didemnin binding pocket prevents ternatin photo-affinity labeling of eEF1A and elicits partial resistance to the antiproliferative effects of didemnin, as well as nearly complete ternatin-4 resistance (*Carelli et al., 2015*; *Krastel et al., 2015*). We thus set out to substantiate eEF1A as the target of ternatin-4 by performing analogous smFRET experiments using recombinantly expressed, human eEF1A(A399V) (*Figure 1—figure supplement 3* and Materials and methods). Following a 2-min incubation with ribosomes in the absence of inhibitor, GTP-bound recombinant wild-type and A399V eEF1A both delivered fluorescently labeled Phe-tRNA$^{Phe}$ to initiation complexes, promoting formation of the pre-translocation complex in which the adjacently bound deacyl- and peptidyl- tRNAs within the ribosome spontaneously and reversibly transit classical (P/P, A/A:~0.7 FRET) and hybrid state positions (P/E, A/A:~0.25 FRET; P/E, A/P:~0.4 FRET; *Figure 1A and G*; *Budkevich et al., 2011*; *Ferguson et al., 2015*). As expected, both didemnin and ternatin-4 effectively prevented pre-translocation complex formation by wild-type eEF1A, yielding a long-lived GA-like state (*Figure 1H1*, *left panels*). By contrast, neither inhibitor (20 µM) had discernible effects on pre-translocation complex formation when eEF1A(A399V) was employed (*Figure 1H1*, *right panels*). These observations validate that the A399V mutation confers didemnin and ternatin-4 resistance during aa-tRNA selection, likely by weakening or occluding small-molecule interactions with eEF1A DIII.

## Elongation complexes trapped by didemnin and ternatin-4 exhibit distinct dynamics

As observed for drugs that target EF-Tu during bacterial aa-tRNA selection (*Geggier et al., 2010*; *Morse et al., 2020*), examination of individual smFRET traces revealed that the GA-like intermediate state captured by didemnin or ternatin-4 exhibited brief transient excursions to both low- (CR-like) and high- (AC-like) FRET states (*Figure 1C and D*). In accordance with the kinetic model of aa-tRNA selection defined in bacteria (*Geggier et al., 2010*; *Morse et al., 2020*), these excursions correspond to initial selection and proofreading events, respectively. To gain insights into the mechanistic impacts of didemnin and ternatin-4 on transitions between states on the tRNA selection reaction coordinate, we analyzed the individual smFRET traces from the pre-steady-state aa-tRNA selection studies above

using hidden Markov modeling (*McKinney et al., 2006*; *Munro et al., 2007*; *Qin, 2004*). Using this approach, we aimed to determine inhibitor-specific differences in the occupancy and kinetic properties of excursions to higher, transient FRET state, which are interpreted as attempts at accommodation (*McKinney et al., 2006*; *Munro et al., 2007*; *Qin, 2004*).

We first assessed conformational dynamics at the ensemble level by compiling transition density plots, in which the FRET values from each single-molecule trajectory immediately before and after a specific FRET transition are revealed as well as the relative transition frequency in and out of each state (*Figure 2A*; *McKinney et al., 2006*). Comparison of transition density plots in the presence of didemnin and ternatin-4 revealed that both inhibitors specifically reduced the frequency of higher-FRET transitions that normally accompany the aa-tRNA selection process by 5–10-fold (*Figure 2A–C* and *Table 1*). Therefore, we focused specifically on AC-like intermediate states sampled prior to the first evidence of high FRET, which were defined as lasting ≥150ms (Materials and methods). Notably, excursions to high-FRET AC-like states were more frequent in the presence of saturating ternatin-4 than didemnin (*Figure 2B–D*). This distinction paralleled a reduction in the overall GA-state lifetime in the presence of saturating ternatin-4 compared to didemnin (*Figure 2E* and *Table 1*).

To discern whether the excursions to high-FRET AC-like states are representative of on-pathway intermediates of the aa-tRNA selection reaction coordinate, we determined the rate at which individual aa-tRNAs eventually formed pre-translocation complexes in the presence of saturating didemnin or ternatin-4. These aa-tRNA selection studies were performed at a lower frame rate (1 Hz) to reduce photobleaching and in the presence of cycloheximide (CHX, 350 µM), which binds in the large subunit exit (E) site to compete with the CCA-end of deacyl-tRNA (*Garreau de Loubresse et al., 2014*). Inclusion of CHX depopulates hybrid tRNA positions (*Ferguson et al., 2015*), which have overlapping FRET efficiencies with tRNAs in mid-FRET GA-like conformations and can complicate analyses of the apparent didemnin- and ternatin-4-stabilized FRET lifetimes (*Figure 2A*). Under both conditions, we observed eventual aa-tRNA accommodation events, albeit extremely slowly (*Figure 2F and G*). We also found that aa-tRNA fully accommodated ~8.5-times faster in the presence of saturating ternatin-4 concentrations compared to saturating didemnin concentrations ($6\times10^{-4}$ s$^{-1}$ vs. $7\times10^{-5}$ s$^{-1}$, respectively) (*Figure 2F and G* and *Figure 2—figure supplement 1A*). These results are consistent with the model that high-FRET, AC-like state excursions on the human ribosome represent transient, on-pathway intermediates in the selection process. We correspondingly infer that ternatin-4 is less efficient than didemnin at inhibiting these excursions and, by extension, at inhibiting aa-tRNA accommodation.

We next sought to distinguish whether the observed excursions to high-FRET AC-like states reflect differences in drug dissociation kinetics or differences in eEF1A dynamics while the drugs remain bound. To do so, we measured the rate of aa-tRNA accommodation from inhibitor-stalled mid-FRET GA-like states following rapid drug washout from the imaging chamber. Here, inhibitor dissociation from eEF1A is expected to enable rapid aa-tRNA accommodation, resulting in a CHX-stabilized, high-FRET classical pre-translocation complex. We found that didemnin dissociates ~25-fold slower than ternatin-4 from stalled elongation complexes (~$2 \times 10^{-4}$ s$^{-1}$ vs ~$5 \times 10^{-3}$ s$^{-1}$, respectively: *Figure 2H1* and *Figure 2—figure supplement 1B-E*). Notably, the apparent drug dissociation rates were >30-fold lower than the frequency of eEF1A conformational changes that allow aa-tRNA to sample AC-like states in the presence of saturating inhibitor (*Figure 2A–C* and *Table 1*). We surmise from these washout studies that excursions to high-FRET, AC-like states in the presence of inhibitor occur while didemnin and ternatin-4 remain bound to eEF1A on the ribosome and represent on-pathway intermediates of tRNA selection. We further posit that these AC-like excursions of aa-tRNA towards the peptidyl transferase center are differentially inhibited by didemnin and ternatin-4 binding.

## Ternatin-4 targets the same binding site as didemnin on eEF1A

To compare the didemnin and ternatin-4 binding sites on eEF1A, we employed ternatin-4 in procedures analogous to those used to solve a didemnin-stalled aa-tRNA selection intermediate by cryo-EM (*Shao et al., 2016*). Ternatin-4 was added to rabbit reticulocyte lysate followed by immunoprecipitation to pull down actively translating ribosomes via the nascent peptide (Materials and methods; *Shao et al., 2016*). These efforts yielded a cryo-EM reconstruction of a ternatin-4/eEF1A/ribosome complex that resolved to 4.1 Å (*Figure 3—figure supplement 1* and *Table 2*). Global features of this map and the position of ternary complex within the ribosomal A site were highly similar to the structure stabilized by didemnin (*Figure 3A*), although the density for aa-tRNA and eEF1A was less

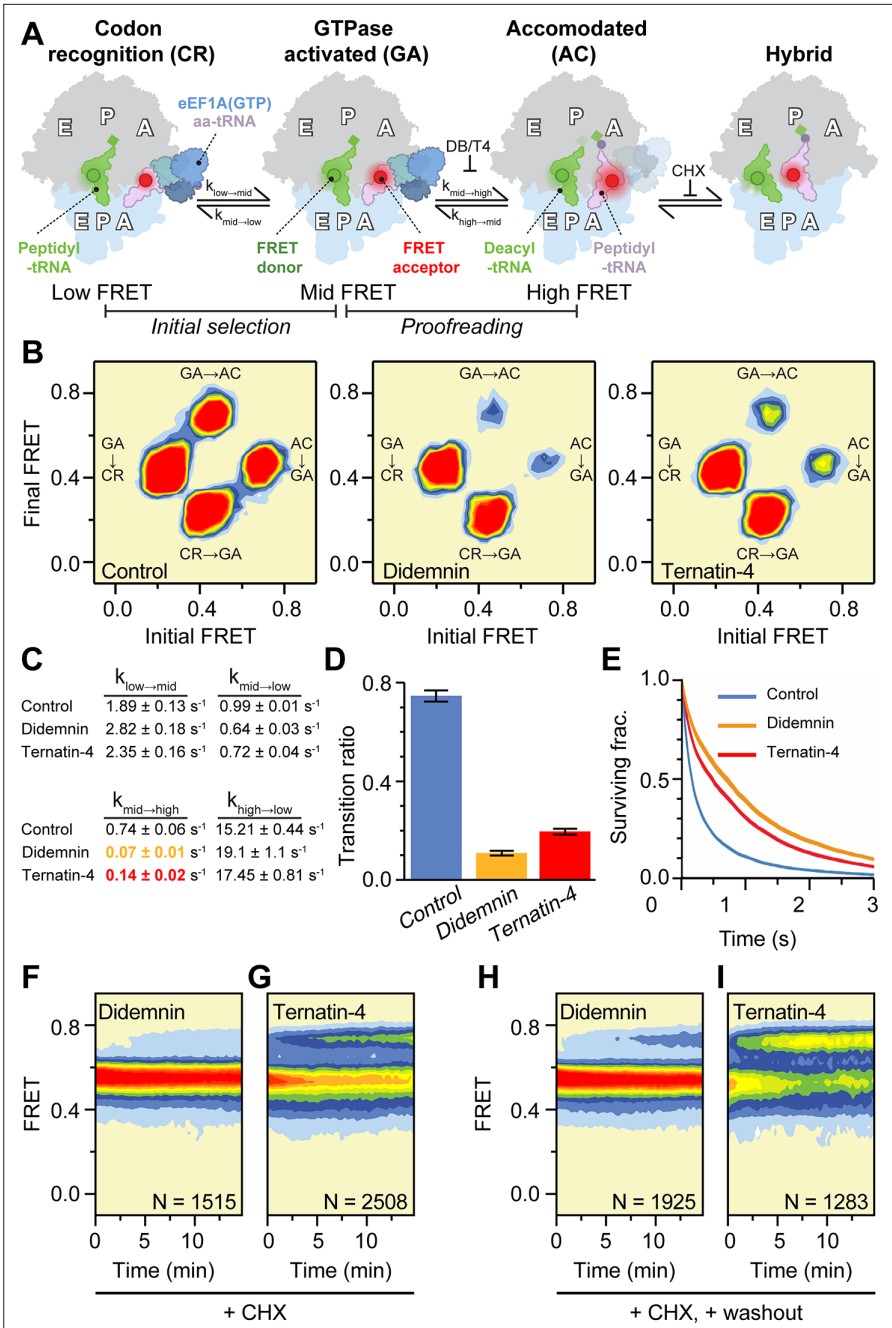

**Figure 2.** Mechanistic differences between didemnin and ternatin-4. (**A**) Schematic of the experimental setup, as in **Figure 1A**. Acceptor Cy5 (red circle)-labeled eEF1A(GTP)-aa-tRNA ternary complex is delivered to 80 S initiation complexes with donor Cy3 (green circle)-labeled P-site tRNA. Hidden Markov modeling (**McKinney et al., 2006**) was used to assess the frequency of transitions from the mid-FRET GTPase-activated (GA,~0.45 FRET) state to low-FRET codon recognition (CR-like,~0.2 FRET) and high-FRET accommodated (AC-like,~0.7 FRET) states, representing "initial selection" and "proofreading", respectively. Transitions from classical high-FRET AC-like states to mid-FRET hybrid states were inhibited by inclusion of cycloheximide (CHX) to aid analysis. (**B**) Transition density plots of pre-accommodated complexes reveal attenuated sampling of the high-FRET state comparing absence of drug (*left*), or in the presence of saturating (20 μM) didemnin (*middle*) or ternatin-4 (*right*). (**C**) Apparent transition rates $k_{i \to j}$ between FRET states (low, mid, high) in aa-tRNA selection experiments prior to the first dwell in high FRET for ≥150ms with standard errors from 1000 bootstrap samples. See also **Table 1**. (**D**) Ratio of mid-to-high (GA-to-AC) over mid-to-low (GA-to-CR) transitions. Error bars: s.e.m. from 1000 bootstrap samples. (**E**) Survival plots reveal increased mid-FRET (GA/GA-like) lifetimes with didemnin and ternatin-4 (line width = s.e.m. from

*Figure 2 continued on next page*

*Figure 2 continued*

1000 bootstrap samples). (**F, G**) Population histograms of N traces after extended incubation in the presence of CHX and 20 μM (**F**) didemnin or (**G**) ternatin-4 reveals 'sneak-through' to high-FRET pre-translocation complexes. (**H**) Didemnin/CHX- or (**I**) ternatin-4/CHX-stalled complexes were washed in the presence of CHX in the first second of each movie, revealing aa-tRNA accommodation, concomitant with drug dissociation. See also *Figure 2—figure supplement 1*.

The online version of this article includes the following figure supplement(s) for figure 2:

**Figure supplement 1.** Didemnin more effective inhibits aminoacyl-tRNA accommodation into the ribosome.

well resolved (*Figure 3—figure supplement 1C*), consistent with ternatin-4 allowing increased ternary complex mobility.

In the presence of ternatin-4, the ribosome adopted an unrotated conformation, with the aa-tRNA in a GA-like conformation within the ribosomal A site and eEF1A bound to the GTPase activating center of the large subunit (*Figure 3A*). Superposition of the cryo-EM map from the ternatin-4-stalled complex with the atomic model of the didemnin-stalled complex (*Shao et al., 2016*) revealed clear density in the cleft between eEF1A DI (G domain) and DIII, which we interpreted as ternatin-4 (*Figure 3A*, *right*). The position of this density overlapped with the hydrophobic didemnin binding site near Ala399 (*Figure 3B–D*), consistent with the A399V resistance mutation disrupting the drug binding pocket via steric clash.

The computationally derived binding pose of ternatin-4 on eEF1A (*Sánchez-Murcia et al., 2017*) could be fit into our experimental density (*Figure 3B* and *Figure 3—figure supplement 2*), though unambiguous assignment of its positioning was limited by local resolution of the cryo-EM map. Comparison of the cryo-EM densities assigned to didemnin and ternatin-4 revealed a larger contact surface for didemnin, consistent with its larger size (*Figure 3* and *Figure 1—figure supplement 1*). Didemnin's structural fold and side chain features also appeared to bridge the gap between the G domain and DIII more effectively than ternatin-4 (*Figure 3*). We note in particular that ternatin-4 does not possess a dimethyltyrosine (Me$_2$Tyr) moiety, which in the didemnin complex is observed to wedge between the G domain and DIII towards the ribosome and nucleotide binding pocket (*Figure 3C and D*).

To connect structural features with the inhibition mechanisms of didemnin and ternatin-4, we reconstituted human 80 S initiation complexes using reagents and procedures analogous to those used for our smFRET studies for cryo-EM analysis. Guided by our smFRET experiments, ternary

**Table 1.** Ternary complex dynamics measured by smFRET.

Apparent transition rates $k_{i \rightarrow j}$ between FRET states (low FRET: index 1; intermediate FRET: index 2; high FRET: index 3) and overall decay rates $k_i$ for each state observed in aa-tRNA selection experiments prior to the first dwell in high FRET (AC) for ≥150ms with standard errors from 1000 bootstrap samples.

**Control**

| | | |
|---|---|---|
| $k_1 = (1.90 \pm 0.13)$ s$^{-1}$ | $k_{1 \rightarrow 2} = (1.89 \pm 0.13)$ s$^{-1}$ | $k_{1 \rightarrow 3} = (0.01 \pm 0.005)$ s$^{-1}$ |
| $k_{2 \rightarrow 1} = (0.99 \pm 0.01)$ s$^{-1}$ | $k_2 = (1.73 \pm 0.09)$ s$^{-1}$ | $k_{2 \rightarrow 3} = (0.74 \pm 0.06)$ s$^{-1}$ |
| $k_{3 \rightarrow 1} = (0.24 \pm 0.09)$ s$^{-1}$ | $k_{3 \rightarrow 2} = (15.21 \pm 0.44)$ s$^{-1}$ | $k_3 = (15.45 \pm 0.43)$ s$^{-1}$ |

**Didemnin**

| | | |
|---|---|---|
| $k_1 = (2.827 \pm 0.18)$ s$^{-1}$ | $k_{1 \rightarrow 2} = (2.824 \pm 0.18)$ s$^{-1}$ | $k_{1 \rightarrow 3} = (0.003 \pm 0.003)$ s$^{-1}$ |
| $k_{2 \rightarrow 1} = (0.64 \pm 0.03)$ s$^{-1}$ | $k_2 = (0.71 \pm 0.03)$ s$^{-1}$ | $k_{2 \rightarrow 3} = (0.07 \pm 0.01)$ s$^{-1}$ |
| $k_{3 \rightarrow 1} = (0.22 \pm 0.17)$ s$^{-1}$ | $k_{3 \rightarrow 2} = (19.1 \pm 1.1)$ s$^{-1}$ | $k_3 = (19.3 \pm 1.1)$ s$^{-1}$ |

**Ternatin-4**

| | | |
|---|---|---|
| $k_1 = (2.36 \pm 0.15)$ s$^{-1}$ | $k_{1 \rightarrow 2} = (2.35 \pm 0.16)$ s$^{-1}$ | $k_{1 \rightarrow 3} = (0.01 \pm 0.01)$ s$^{-1}$ |
| $k_{2 \rightarrow 1} = (0.71 \pm 0.04)$ s$^{-1}$ | $k_2 = (0.85 \pm 0.04)$ s$^{-1}$ | $k_{2 \rightarrow 3} = (0.14 \pm 0.02)$ s$^{-1}$ |
| $k_{3 \rightarrow 1} = (0.52 \pm 0.42)$ s$^{-1}$ | $k_{3 \rightarrow 2} = (17.45 \pm 0.81)$ s$^{-1}$ | $k_3 = (17.97 \pm 0.83)$ s$^{-1}$ |

**Table 2.** Cryo-EM data collection and processing statistics.

| | RRL 80 S•aa-tRNA• | Human 80 S• Phe-tRNA[Phe]• | Human 80 S• Phe-tRNA[Phe]• |
|---|---|---|---|
| | eEF1A•ternatin-4 | eEF1A•didemnin | eEF1A•ternatin-4 |
| | EMDB-27732 | EMDB-27691 | EMDB-27694 |
| **Grid Preparation** | | | |
| Grids | Quantifoil R2/2 +5 nm carbon | UltrAuFoil Gold R1.2/1.3 | UltrAuFoil Gold R1.2/1.3 |
| [Drug] (μM) | 1 | 0.2 | 20 |
| [80 S] (nM) | ~120 | ~200 | ~200 |
| Plunge freezer | Vitrobot MKIII (FEI) | Vitrobot MKII (FEI) | Vitrobot MKII (FEI) |
| Temperature (°C) | 4 | 4 | 4 |
| Humidity (%) | 100 | 100 | 100 |
| Wait time (s) | 30 | 45 | 45 |
| Blot time (s) | 3 | 2–3 | 2–3 |
| **Data Collection** | | | |
| Microscope | Titan Krios (FEI) | Titan Krios (FEI) | Titan Krios (FEI) |
| Voltage (kV) | 300 | 300 | 300 |
| Camera | Falcon II (FEI) | K2 Summit (Gatan) | K2 Summit (Gatan) |
| Acquisition software | EPU (FEI) | Leginon MSI | Leginon MSI |
| Acquisition mode | Counting | Counting | Super Resolution |
| Magnification (×) | 135,000 | 105,000 | 105,000 |
| Pixel size (Å) | 1.040 | 1.073 | 0.548 (1.096) |
| Calibrated pixel (Å) | 1.045 | 1.067 | 1.085 |
| Defocus range (μm) | 0.8–9.2 | 0.8–3.2 | 0.8–2.5 |
| Frames/movie | 40 | 50 | 50 |
| Exposure time (s) | 1 | 10 | 10 |
| Dose rate ($e^-/Å^{-2}/s$) | 40 | 6.72 | 6.53–7.00 |
| Frame rate (s/frame) | 0.06 | 0.2 | 0.2 |
| Electron dose ($e^-/Å^{-2}$) | 40.00 | 67.19 | 65.31–70.00 |
| **Data Processing** | | | |
| Useable micrographs | 1,422 | 3,574 | 9,401 |
| Particles picked/sorted | 159,068 | 301,208 | 934,291 |
| Particles after 2D | 120,986 | 289345 | 816,898 |
| Particles after 3D | 85,795 | 180,555 | 393,664 |
| Final particles | 22,034 | 35,193 | 34,369 |
| Sharpening B-factor ($Å^2$) | −110.3 | −20 | −20 |
| Resolution (Å) | 4.1 | 3.6 | 3.2 |

complex was delivered to human initiation complexes in the presence of either didemnin (200 nM) or ternatin-4 (20 μM) and flash frozen on cryo-EM grids within 2 min. The resulting structures of human tRNA selection intermediates were notably similar to those isolated from rabbit reticulocyte lysate. Overall, the human structures resolved to higher resolution (3.2–3.6 Å) and, due to reconstitution with

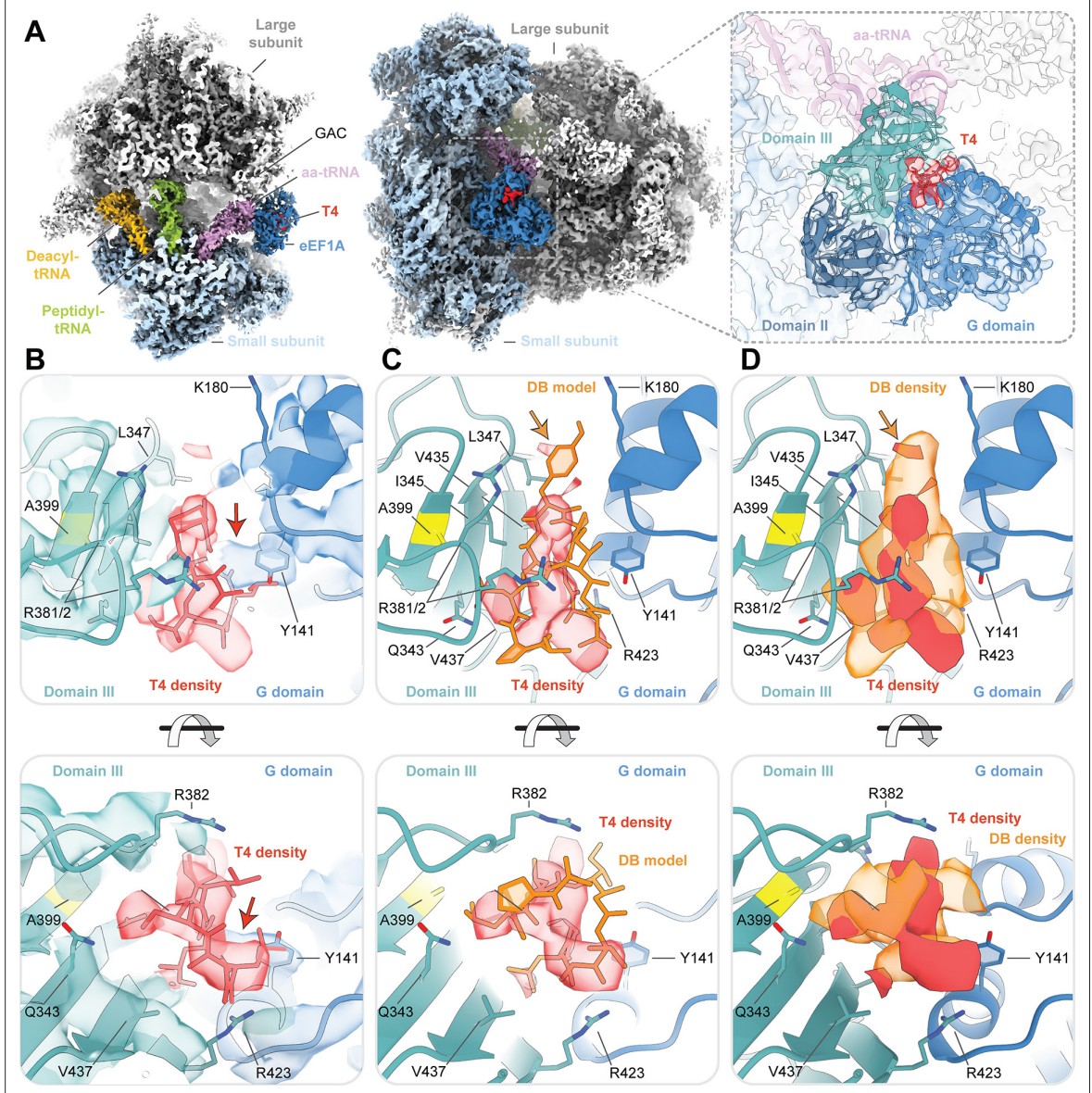

**Figure 3.** Cryo-EM structure of ternatin-4 stalled rabbit 80S-eEF1A-aa-tRNA complex. (**A**) Overview of the cryo-EM density map of the ternatin-4-stalled rabbit elongation complex viewed from the small subunit head domain (*left*) and into the GTPase activating center (GAC) from the leading edge (*middle*) comprising the large (gray) and small (light blue) ribosomal subunits, peptidyl-tRNA (P site, green), deacyl-tRNA (E site, gold), aminoacyl-tRNA in the pre-accommodated 'A/T' state (aa-tRNA, purple), eEF1A (blue), and ternatin-4 (T4, red). Inset (*right*) shows cryo-EM density of eEF1A ternary complex on the ribosome highlighting density for T4 with eEF1A colored by domain. Atomic model of eEF1A and aa-tRNA from PDB-ID: 5LZS (*Shao et al., 2016*) was rigid-body fit into the cryo-EM map. (**B**) Zoom-in of cryo-EM density at the interface between the G domain and domain III of eEF1A that has been assigned to T4, colored as in (A, *right*). Residue A399 (yellow), which confers resistance to didemnin (DB) and T4 and when mutated to valine, is adjacent to the density for T4. Computationally derived model of T4 (*Sánchez-Murcia et al., 2017*) was rigid-body fit into the cryo-EM density. Red arrow denotes unassigned density that could be assigned to T4. (**C, D**) Overlay of (**C**) the atomic model and (**D**) the cryo-EM density of DB (orange, EMD-4130) and T4, Orange arrow denotes the dimethyltyrosine (Me$_2$Tyr) moiety of didemnin. All cryo-EM density is contoured at 3.5 σ. See also *Figure 3—figure supplements 1–4*.

The online version of this article includes the following figure supplement(s) for figure 3:

**Figure supplement 1.** Cryo-EM processing of the ternatin-4 stalled rabbit 80S-eEF1A-aa-tRNA structure.

**Figure supplement 2.** Structural comparison of the ternatin-4-stalled rabbit elongation complex with a published prediction of ternatin binding.

**Figure supplement 3.** Cryo-EM processing of the didemnin and ternatin-4 stalled human 80S-eEF1A-aa-tRNA structures.

**Figure supplement 4.** Cryo-EM structures of didemnin and ternatin-4 stalled human 80S-eEF1A(aa-tRNA) complexes.

defined tRNAs, possessed better quality cryo-EM density for the A- and P-site tRNAs (*Figure 3—figure supplement 3* and *Table 2*).

As was observed above in the structures captured from rabbit reticulocyte lysate, the cryo-EM density for eEF1A and aa-tRNA was better resolved in the structure stalled by didemnin than that stalled by ternatin-4 (*Figure 3—figure supplement 3D, E*). Consistent with the differences in ternary complex dynamics revealed by smFRET, the human sample stalled by ternatin-4 required nearly three times the number of 80 S particles as the sample stalled by didemnin to yield a final structure with well-defined cryo-EM density for eEF1A (*Figure 3—figure supplement 3*). The structures derived from reconstituted human reagents corroborated the binding sites observed in rabbit, supporting a larger relative surface area covered by didemnin than ternatin-4 imparted largely by the Me$_2$Tyr moiety of didemnin (*Figure 3—figure supplement 4*).

Collectively, these structural results show that didemnin and ternatin-4 share the same binding site on eEF1A, suggesting that both inhibitors restrict conformational changes in eEF1A through their binding to the DI-DIII interface. We further propose that the smaller footprint of ternatin-4 on eEF1A compared to didemnin contributes to the differences in dynamics evidenced in both drug-bound tRNA selection intermediates by smFRET.

## Ternatin-4 traps eEF1A on the ribosome with disordered switch loops

To understand the differential effects of didemnin and ternatin-4 on eEF1A GTPase function, we next compared the conformation of the G domain (DI) of eEF1A in the didemnin vs. ternatin-4-bound rabbit 80 S complex. As was observed in the didemnin-stalled intermediate (*Shao et al., 2016*), eEF1A bound to ternatin-4 revealed no density for a terminal (γ) phosphate in the nucleotide binding pocket (*Figure 4A*), consistent with ternatin-4 stalling of eEF1A after GTP hydrolysis and P$_i$ release. However, we observed differences in the stability of the switch loops of eEF1A's G domain when stalled by didemnin as compared to ternatin-4 (*Figure 4B*)—elements that canonically become disordered following GTP hydrolysis and P$_i$ release in Ras-family GTPases (*Bourne et al., 1991*; *Gasper and Wittinghofer, 2019*). In the ternatin-4-bound rabbit 80 S complex, the switch-I and -II elements of eEF1A displayed particularly weak cryo-EM density (*Figure 4B*). By contrast, the previously determined cryo-EM structure of the didemnin-trapped rabbit 80 S ribosome (*Shao et al., 2016*) exhibited comparatively ordered cryo-EM density for both switch loops despite P$_i$ dissociation (*Figure 4B*). This was also true of the didemnin-bound human 80 S ribosome reported here (*Figure 4—figure supplement 1*).

Comparatively, the ternatin-4-stalled structures displayed weakened density for the putative catalytic His95 of switch II, for the hydrophobic gate elements of the P loop (Val16) and switch I (Ile71), and for the C-terminal helix (α4) of switch I (*Figure 4A and B* and *Figure 4—figure supplement 1A, B*). Further, we observed a loss of switch I residue Arg69 contact with the aa-tRNA minor groove (*Figure 4B* and *Figure 4—figure supplement 1B*). We did, however, observe maintenance of the switch-II hydrophobic gate residue Phe98 and possibly strengthened contact between switch-I residue Lys51 and the large subunit (*Figure 4D* and *Figure 4—figure supplement 1C*), potentially serving to stabilize eEF1A on the large subunit and prevent aa-tRNA accommodation. Small-subunit contact with N-terminal Arg37 of the α2 helix in eEF1A was evident in all structures, but we observed weakened density for the C-terminal portion of the α2 helix in the ternatin-4-stalled complexes (*Figure 4B and C* and *Figure 4—figure supplement 1B, C*). In the ternatin-4-stalled rabbit ribosome structure, contact between the α2 helix and the large subunit appeared weakened compared to the didemnin-stalled structure (*Figure 4C*), though this trend was less pronounced in the human structures (*Figure 4—figure supplement 1C*).

These findings indicate that, like didemnin, ternatin-4 stalls eEF1A on the ribosome after GTP hydrolysis and P$_i$ release. However, while didemnin stalls eEF1A in an active, GTP-bound conformation, switch elements within the G domain of the ternatin-4 stalled complex are more disordered, suggesting increased post-hydrolysis dynamics in the ternatin-4-stalled state.

## Protein synthesis inhibition by ternatin-4, but not didemnin, can be reversed in cells

The observed differences in the structure and dynamics of didemnin- and ternatin-4-stalled eEF1A on the ribosome prompted us to investigate whether similar kinetic differences could be discerned

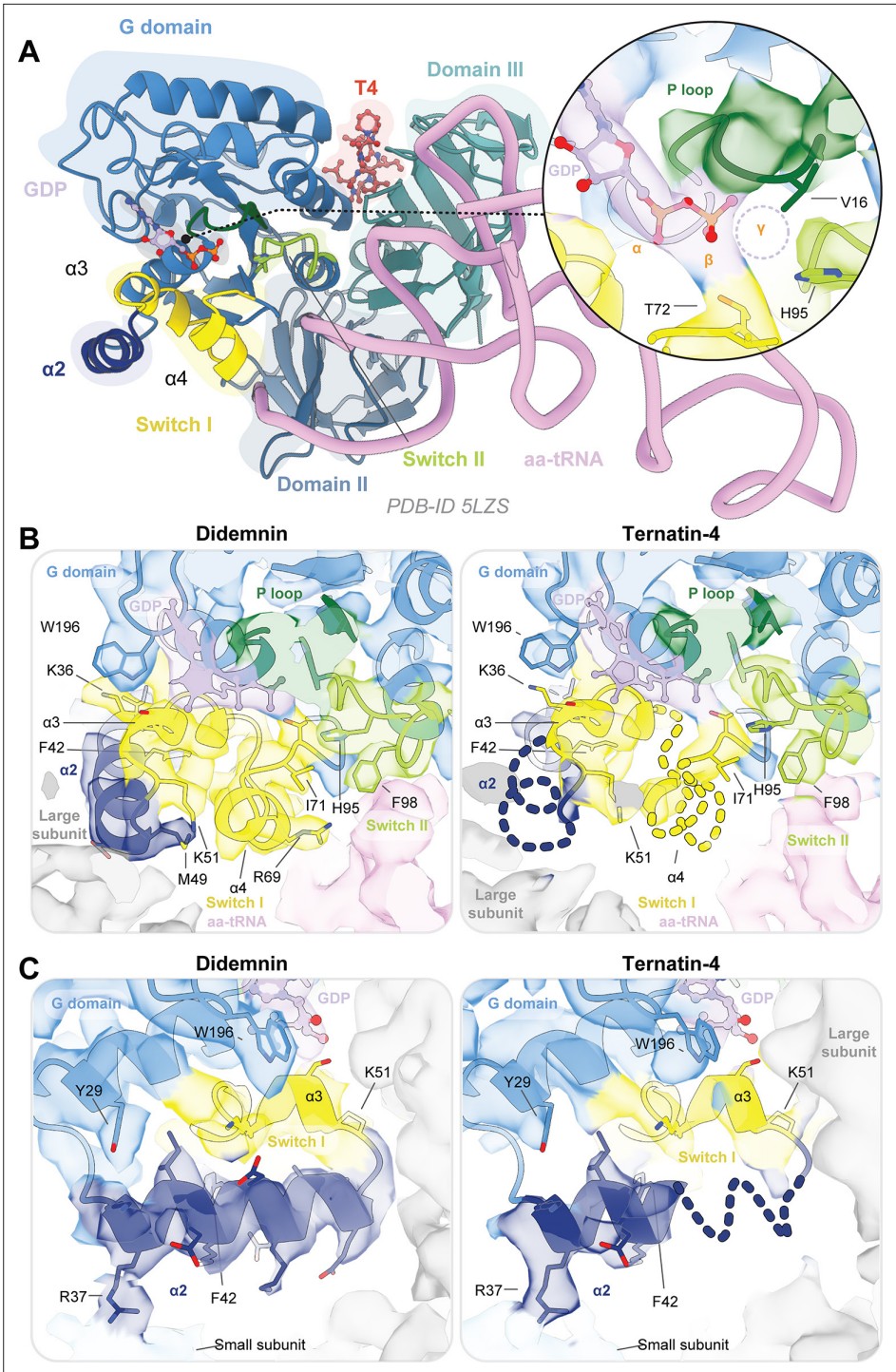

**Figure 4.** eEF1A is more dynamic when bound to ternatin-4 than to didemnin. (**A**) Overview of the domain architecture of the eEF1A ternary complex from PDB-ID: 5LZS (*Sánchez-Murcia et al., 2017*) as viewed from the leading edge of the rabbit 80 S ribosome and the sarcin ricin loop. G-domain (blue) elements include switch I (yellow), switch II (lime), the P loop (dark green), helix α2 (dark blue), and a bound GDP (light purple) in the nucleotide binding pocket. Inset shows density for the GDP nucleotide. Model of the computationally derived model of ternatin-4 (T4, red) (*Shao et al., 2016*) is shown to indicate the drug binding pocket location. (*Inlay*) Dotted circle indicates the approximate location of the γ phosphate when GTP is bound. (**B, C**) Atomic models of eEF1A and aa-tRNA from PDB-ID: 5LZS were rigid-body fit into the cryo-EM density shown in surface representation of the eEF1A G domain when stalled with didemnin (DB, orange, EMD-4130, *left*) (*Shao et al.,*

*Figure 4 continued on next page*

*Figure 4 continued*

**2016**) or T4 (*right*) on the elongating rabbit 80 S ribosome, colored as in (**A**). Dotted lines indicate regions of weakened cryo-EM density in the T4-stalled eEF1A G domain in the C terminus of helix α2. Panels highlight (**B**) the nucleotide binding pocket of eEF1A and switch loop architecture and (**C**) the junction between the large subunit, eEF1A helix α2, and small subunit (light blue) rRNA helix 14. All cryo-EM density is contoured at 3 σ. See also *Figure 4—figure supplement 1*.

The online version of this article includes the following figure supplement(s) for figure 4:

**Figure supplement 1.** The G domain of eEF1A is more dynamic when bound to ternatin-4 than to didemnin on the human 80 S ribosome.

---

in cells. After 4 hr of continuous treatment, both compounds potently inhibited protein synthesis in HCT116 cells (didemnin $IC_{50}$ ~7 nM; ternatin-4 $IC_{50}$ ~36 nM; *Figure 5A*), as measured by metabolic labeling with homopropargylglycine (Hpg) and flow cytometry analysis (*Beatty et al., 2006*). Cells treated with ternatin-4 (500 nM, ~14 × $IC_{50}$), followed by rigorous washout, recovered protein synthesis rates to ~25% of starting levels within 22 hr (*Figure 5B and C*). By contrast, protein synthesis was undetectable for at least 22 hr followed by washout in cells treated with saturating didemnin (100 nM, ~14 × $IC_{50}$).

To substantiate these kinetic differences, we monitored time-dependent induction of apoptosis under conditions of continuous drug exposure or after a brief pulse, followed by washout. We used Jurkat cells which are known to undergo rapid apoptosis in the presence of didemnin (*Baker et al., 2002*). Continuous exposure to didemnin or ternatin-4 for 24 hr induced apoptosis in >95% of the cells, with didemnin being ~sevenfold more potent than ternatin-4 (didemnin $IC_{50}$ ~4 nM; ternatin-4 $IC_{50}$ ~30 nM; *Figure 5—figure supplement 1*). Treatment with saturating didemnin or ternatin-4 induced membrane phosphatidylserine exposure (an early marker of apoptosis) within 2–4 hr in a subpopulation of cells (*Figure 5D*). A 2 hour pulse with saturating ternatin-4 was sufficient to induce apoptosis in ~40% of the cells, whereas rigorous washout followed by 22 hr incubation in compound-free media resulted in no further cell death (*Figure 5E*). By contrast, cell death increased from ~20% after a 2 hr didemnin pulse to ~75% after drug washout, consistent with didemnin's ability to inhibit protein synthesis in a sustained, washout-resistant manner (*Figure 5C*). Collectively, these results demonstrate clear differences in cellular pharmacology between didemnin and ternatin-4 under conditions of transient drug exposure followed by washout, which correlate with the ~25-fold higher dissociation rate of ternatin-4 observed by smFRET (*Figure 2*).

## Discussion

In this study, we combined the complementary methods of smFRET, cryo-EM, and in vivo translation measurements to reveal the molecular mechanisms by which didemnin and ternatin-4 inhibit translation in mammals. Both didemnin or ternatin-4 bind similar sites at the eEF1A DI-DIII interface, which likely prevents the inter-domain rearrangements that allow for aa-tRNA accommodation into the ribosomal A site and eEF1A dissociation from the ribosome. Consequently, didemnin and ternatin-4 trap eEF1A(GDP)aa-tRNA ternary complex for extended periods during an intermediate stage of the aa-tRNA selection process, which was observed here by smFRET and cryo-EM.

The finding that both structurally and chemically distinct molecules trap aa-tRNA in GA-like states argues that they similarly inhibit rate-limiting conformational changes within eEF1A(GDP)-aa-tRNA ternary complex during the proofreading stage of tRNA selection that allow aa-tRNA release and accommodation into the ribosomal A site. Individual smFRET traces revealed that the GTPase activated-like intermediate state captured by didemnin or ternatin-4 exhibited transient excursions to both lower- (CR-like) and higher- (AC-like) FRET states, though transitions to higher-FRET states were primarily affected by both compounds. This pattern of inhibition is consistent with smFRET studies in the presence of drugs that target EF-Tu, the bacterial homolog of eEF1A, during aa-tRNA selection (*Geggier et al., 2010*; *Morse et al., 2020*). By inference from mechanistic investigations of aa-tRNA selection on the bacterial ribosome (*Geggier et al., 2010*; *Morse et al., 2020*), and the cryo-EM structures presented here, we propose that didemnin and ternatin-family cyclic peptides target the proofreading stage of aa-tRNA selection by inhibiting essential dynamics within eEF1A that facilitate aa-tRNA accommodation after GTP hydrolysis. Further, we posit that AC-like excursions of aa-tRNA

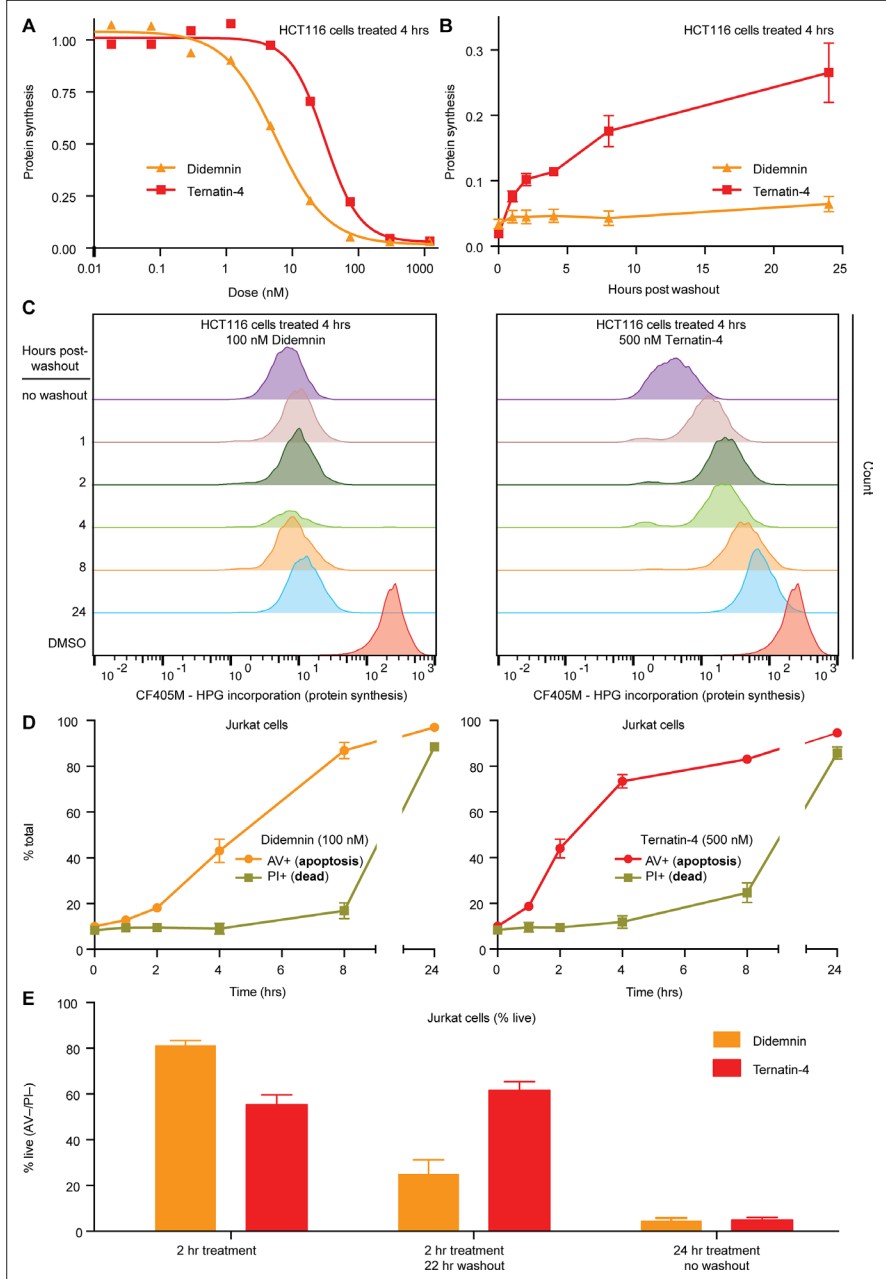

**Figure 5.** Cellular effects of ternatin-4, but not didemnin, are reversible upon washout. (**A**) Dose-dependent effects of didemnin (orange) and ternatin-4 (red) on protein synthesis in HCT116 cells under continuous treatment (4 hr). Protein synthesis was quantified by homopropargylglycine pulse (1 hr) followed by fixation and copper-mediated conjugation to CF405M azide fluorophore and analyzed by FACS. (**B**) HCT116 cells were treated with didemnin (100 nM, orange) or ternatin-4 (500 nM, red) for 4 hr, followed by washout. Protein synthesis was quantified as in (**A**) at 1, 2-, 4-, 8-, or 24 hr post-washout. (**C**) Histograms corresponding to panel (**B**) for didemnin (*left*) and ternatin-4 (*right*). (**D**) Jurkat cells were treated with didemnin (100 nM, *left*) or ternatin-4 (500 nM, *right*) or for 1, 2, 4, 8, or 24 hr, stained with Annexin V-FITC (AV+, apoptotic) and propidium iodide (PI+, green, dead), and analyzed by FACS. (**E**) Jurkat cells were treated with didemnin (100 nM, orange) or ternatin-4 (500 nM, red) for 2 hr followed by washout and 22 hr incubation in drug-free media or for 24 hr and analyzed as in (**D**). Bar graphs show the ratio of live cells. See also *Figure 5—figure supplement 1*.

The online version of this article includes the following figure supplement(s) for figure 5:

**Figure supplement 1.** Dose response curve for Jurkat cells treated for 24 hr with the indicated compound.

towards the peptidyl transferase center are coupled to conformational changes in eEF1A, which are differentially inhibited by didemnin and ternatin-4 binding.

Compared to didemnin, ternatin-4 traps eEF1A in a more dynamic state, characterized by conformational heterogeneities seen by smFRET and a reduced extent of eEF1A/ribosome and /aa-tRNA contacts observed by cryo-EM. In the ternatin-4-stalled intermediate structure, we observed particularly weak cryo-EM density for the eEF1A switch-I element and helix α2. This helical insertion, which is not present in the bacterial equivalent of eEF1A, makes direct contact with both the small and large subunits and appears to stabilize the active conformation of switch I. We hypothesize that the observed disordering of aa-tRNA and eEF1A in the ternatin-4-stalled ternary complex reflects an increase in post-GTP hydrolysis eEF1A dynamics as compared to the didemnin-inhibited factor. We note in this context that the increased dynamics evidenced by the cryo-EM structures of ternatin-4-stalled complexes correlated with those evidenced by smFRET within the drug-stalled, GA-like state. These analyses suggest that ternatin-4 is less efficient than didemnin in preventing conformational changes within eEF1A that allow aa-tRNA to enter the peptidyl transferase center. We hypothesize that the differences in dynamics of the didemnin-bound vs. ternatin-4-bound intermediates observed by smFRET could be explained in part by differences in drug size and in differential eEF1A residue engagement.

While didemnin and ternatin-4 exhibit similar biochemical potencies in our in vitro translation reactions, we find that ternatin-4 dissociates ~25 × more rapidly than didemnin under washout conditions. Correspondingly, while didemnin and ternatin-4 inhibit protein synthesis and induce apoptosis at similar concentrations, the cellular effects of didemnin were quasi-irreversible on washout whereas those of ternatin-4 were reversible.

Identifying mammalian translation inhibitors effective in the clinical treatment of disease has proven to be a tall challenge with many obstacles (*Burgers and Fürst, 2021*; *Fan and Sharp, 2021*). The present investigations suggest, however, that natural products and their synthetic variants targeting translation elongation could potentially be tailored to create compounds that are less toxic while maintaining efficacy. As with other systems (*Copeland, 2021*), our findings suggest that the in vivo residence times of didemnin and ternatin-4 likely influence their efficacy and toxicity profiles. Given the rapid response of some cell lines to eEF1A inhibitors, there is a possibility for achieving efficacy against rapidly proliferating cells in vivo with a ternatin-like inhibitor, while sparing the broad toxicity of irreversible translation inhibition. Indeed, we have recently found that a hydroxylated variant of ternatin-4 is efficacious in a mouse model of MYC-dependent B cell lymphoma (*Wang et al., 2022*).

Recent work has shown that ternatin-4 has an $IC_{90}$ of 15 nM against SARS-CoV-2 in vitro (*Gordon et al., 2020*). A related study showed that plitidepsin (also known as dehydrodidemnin B, a close structural relative of didemnin originally advanced in clinical trials for the treatment of multiple myeloma) also possessed potent antiviral activity against SARS-CoV-2 with an $IC_{90}$ of 0.88 nM (*White et al., 2021*). Although preliminary, such findings underscore the potential value of advancing translation elongation-targeting drugs towards human therapeutics. They also motivate investigations seeking to identify the specific molecular context that determines drug sensitivity and the downstream signaling consequences. Progress on this front will be invaluable for identifying unexploited and forthcoming therapeutic applications for inhibitors of eEF1A. Efforts of this kind are also likely to inform on the principles by which small-molecule therapies can exploit and specifically target vulnerabilities of repetitive translation elongation reactions for the treatment of human disease.

## Materials and methods

**Key resources table**

| Reagent type (species) or resource | Designation | Source or reference | Identifiers | Additional information |
|---|---|---|---|---|
| Cell line (*Homo sapiens*) | HCT116 | ATCC | Cat#CCL-247 | |
| Cell line (*Homo sapiens*) | HEK293T | ATCC | Cat#CRL-3216 | Adherent cells |

*Continued on next page*

*Continued*

| Reagent type (species) or resource | Designation | Source or reference | Identifiers | Additional information |
|---|---|---|---|---|
| Cell line (*Homo sapiens*) | Jurkat cells | ATCC | Cat#TIB-152 | |
| Genetic reagent (*Homo sapiens*) | HCT116 cells expressing Flag-eEF1A1(WT)-P2A-mCherry | *Carelli et al., 2015*; DOI:10.7554/eLife.10222 | | Generated using a pHR lentiviral vector |
| Genetic reagent (*Homo sapiens*) | HCT116 cells expressing Flag-eEF1A1(A399V)-P2A-mCherry | *Carelli et al., 2015*; DOI:10.7554/eLife.10222 | | Generated using a pHR lentiviral vector |
| Biological sample (*Oryctolagus cuniculus*) | Rabbit reticulocyte lysate (RRL) | Green Hectares | N/A | https://greenhectares.com |
| Recombinant DNA reagent (*Homo sapiens*) | 3×Flag-tagged KRas transcript | *Shao et al., 2016*; DOI:10.1016/j.cell.2016.10.046 | | SP64-based plasmid purified PCR product 5–20 ng/ml |
| Sequence-based reagent | *MFF* synthetic mRNA | Dharmacon/ Thermo Fisher Scientific *Ferguson et al., 2015* | | CAA CCU AAA ACU UAC ACA CCC UUA GAG GGA CAA UCG AUG UUC AAA GUC UUC AAA GUC AUC |
| Sequence-based reagent | DNA oligo A | Dharmacon/ Thermo Fisher Scientific *Ferguson et al., 2015* | | AAA AAA AAA AAA AAA AAA AAA AAA AAA AAA |
| Sequence-based reagent | DNA oligo B | Dharmacon/ Thermo Fisher Scientific *Ferguson et al., 2015* | | GTA AGT TTT AGG TTG CCC CCC TTT TTT TTT TTT TTT TTT TTT TTT TTT TTT 3'-Biotin-TEG |
| Peptide, recombinant protein | 3×Flag peptide | Sigma-Aldrich | Cat#SAE0194 | |
| Peptide, recombinant protein | Creatine kinase | Roche/ Sigma-Aldrich | Cat#10127566001 | 40 mg/ml |
| Peptide, recombinant protein | Myokinase | Sigma-Aldrich | Cat#M5520 | |
| Peptide, recombinant protein | Pyruvate Kinase | Sigma-Aldrich | Cat#P9136 | |
| Antibody | Anti-EF1α, CBP-KK1 | MilliporeSigma | Cat#05–235 | Purified mouse monoclonal IgG1κ |
| Chemical compound, drug | 3,4-Dihydroxybenzoic acid (PCA) | Sigma-Aldrich | Cat#37580 | |
| Chemical compound, drug | 4-nitrobenzyl alcohol (NBA) | Sigma-Aldrich | Cat#N12821 | |
| Chemical compound, drug | CF405M-azide | Biotium | Cat#92092 | |
| Chemical compound, drug | Creatine phosphate | Roche/ Sigma-Aldrich | Cat#10621714001 | 12 mM |
| Chemical compound, drug | Cycloheximide (CHX) | MP Biomedicals/ Thermo Fisher Scientific | Cat#MP219452701 | |
| Chemical compound, drug | Cyclooctatetraene (COT) | Sigma-Aldrich | Cat#138924 | |
| Chemical compound, drug | Didemnin B | NCI/DTP | CAS 77327-05-0 | |
| Chemical compound, drug | EasyTag L-[$^{35}$S]-methionine | Perkin Elmer | Cat#NEG709A500UC | 0.5 mCi |
| Chemical compound, drug | L-homopropargyl glycine | Kerafast | Cat# FCC438 | |

*Continued on next page*

*Continued*

| Reagent type (species) or resource | Designation | Source or reference | Identifiers | Additional information |
|---|---|---|---|---|
| Chemical compound, drug | Mammalian Protease Arrest protease inhibitor | G-Biosciences | Cat#786–331 | |
| Chemical compound, drug | Phosphoenolpyruvate (PEP) | Sigma-Aldrich | Cat#P7127 | |
| Chemical compound, drug | Protocatechuate 3,4-dioxygenase (PCD) | Sigma-Aldrich | Cat#P8279-25UN | |
| Chemical compound, drug | Puromycin (dihydrochloride) | Sigma-Aldrich | Cat#58-58-2 | |
| Chemical compound, drug | RNasin Plus | Promega | Cat#N2611 | |
| Chemical compound, drug | Ternatin-4 | *Carelli et al., 2015*; DOI:10.7554/eLife.10222 | | |
| Chemical compound, drug | Trolox | Sigma-Aldrich | Cat#238813 | |
| Commercial assay or kit | Annexin V-FITC | BD Pharmingen | Cat#556420 | |
| Commercial assay or kit | Anti-FLAG M2 magnetic beads | Sigma-Aldrich | Cat#A2220 | |
| Commercial assay or kit | MycoAlert Mycoplasma Detection Kit | Lonza | Cat# LT07-701 | |
| Commercial assay or kit | Propidium iodide | BD Pharmingen | Cat#556463 | |
| Commercial assay or kit | Zombie Red | BioLegend | Cat#423109 | |
| Software, algorithm | FlowJo | Tree Star | N/A | https://www.flowjo.com/solutions/flowjo |
| Software, algorithm | CTFFIND4 | *Rohou and Grigorieff, 2015*; DOI:10.1016/j.jsb.2015.08.008 | | http://grigoriefflab.janelia.org/ctffind4 |
| Software, algorithm | Leginon | *Suloway et al., 2005*; DOI:10.1016/j.jsb.2005.03.010 | | http://leginon.org |
| Software, algorithm | MotionCorr | *Li et al., 2013*; DOI:10.1038/nmeth.2472 | | |
| Software, algorithm | RELION | *Kimanius et al., 2016*; DOI:10.7554/eLife.18722 DOI:10.1042/BCJ20210708 | | version 2.0 version 4.0 |
| Software, algorithm | SPARTAN | *Juette et al., 2016*; DOI:10.1038/nmeth.3769 | | version 3.3.0 http://scottcblanchardlab.com/software |
| Software, algorithm | UCSF Chimera | *Pettersen et al., 2004*; DOI:10.1002/jcc.20084 | | |
| Software, algorithm | UCSF ChimeraX | *Pettersen et al., 2021*; DOI:10.1002/pro.3943 | | |

## Ribosome subunit isolation for human smFRET and cryo-EM analysis

Human ribosome subunits were isolated from adherent HEK293T cells as described previously (*Ferguson et al., 2015*). Cells were grown in high-glucose Dulbecco's modified Eagle's medium (DMEM, Life Technologies) supplemented with 10% fetal bovine serum (FBS; Atlanta Biologicals) and 1% penicillin/streptomycin (Life Technologies) and tested for mycoplasma (Lonza) before use. At ~75% confluency, 350 µM cycloheximide (CHX) was added to the medium and incubated for 30 min. Cells were detached with 0.05% Trypsin-EDTA supplemented with 350 µM CHX, pelleted and flash-frozen in liquid nitrogen for storage.

For ribosome isolation, cells were thawed on ice and resuspended in lysis buffer (20 mM Tris HCl pH 7.5, 10 mM KCl, 5 mM MgCl$_2$, 1 mM DTT, 5 mM putrescine, 350 µM CHX, 4 units ml$^{-1}$ RNAse

Out (Thermo Fisher), 1×Halt Protease Inhibitor Cocktail without EDTA (Thermo Fisher)), followed by addition of 0.5% (v/v) NP-40, 0.5% (w/v) sodium deoxycholate, and 20 units ml$^{-1}$ Turbo DNAse (ThermoFisher) and incubation at 4 °C for 20 minutes while rotating. Lysate was clarified by brief centrifugation, loaded onto pre-chilled 10–50% sucrose gradients prepared in polysome gradient buffer (20 mM Tris HCl pH 7.5, 10 mM KCl, 5 mM MgCl$_2$, 1 mM DTT, 5 mM putrescine, 350 μM CHX), and centrifuged for 3 hr at 110 krcf, 4 °C. Gradients were analyzed following standard procedures on a gradient fractionator (Brandel) with UV absorbance detector (Teledyne Isco), polysome fractions were collected and pelleted for 18 hr at 125 krcf, 4 °C. Polysome pellets were rinsed and resuspended in resuspension buffer (20 mM Tris HCl pH 7.5, 50 mM KCl, 1.5 mM MgCl$_2$, 1 mM DTT), followed by addition of 1 mM puromycin, raising the KCl concentration to 500 mM, 30 min incubation at 4 °C with rotation, and 15 min incubation at 37 °C. The solution was cleared by centrifugation, loaded onto 15–30% sucrose gradients prepared in subunit gradient buffer (5 mM Tris HCl pH 7.5, 500 mM KCl, 2.5 mM MgCl$_2$, 1 mM DTT) and centrifuged for 14 hr at 50 krcf, 20 °C. Gradients were analyzed as before and individual small and large subunit fractions were collected, followed by centrifugation for 3 hr (large) or 6 hr (small) at 425 krcf, 4 °C. Subunit pellets were resuspended in 80 S polymix buffer (30 mM HEPES pH 7.5, 5 mM MgCl$_2$, 50 mM NH$_4$Cl, 5 mM Putrescine, 2 mM Spermidine, 1 mM DTT), aliquoted, and flash-frozen in liquid nitrogen.

## Human 80S initiation complex formation for smFRET and cryo-EM

For smFRET analyses, synthetic mRNA (Dharmacon, sequence CAA CCU AAA ACU UAC ACA CCC UUA GAG GGA CAA UCG **AUG UUC AAA** GUC UUC AAA GUC AUC) was prepared for surface immobilization in the following way: 45 μM each of DNA oligonucleotides A (AAA AAA AAA AAA AAA AAA AAA AAA AAA AAA) and B (GTA AGT TTT AGG TTG CCC CCC TTT TTT TTT TTT TTT TTT TTT TTT TTT TTT with 3'-Biotin-TEG modification) in hybridization buffer (10 mM HEPES pH 7, 150 mM KCl, 0.5 mM EDTA) were heated to 95 °C for 5 min and annealed on ice for 5 min. The annealed oligonucleotide (20 μM) and mRNA (20 μM) in hybridization buffer were incubated for 5 minutes at 37 °C and for 5 minutes on ice. For smFRET analyses, tRNA$^{fMet}$ isolated from *E. coli* was labeled with Cy3 at the s$^4$U8 residue following established procedures (*Blanchard et al., 2004b*). Cy3-labeled (smFRET) or unlabeled (cryo-EM) tRNA$^{fMet}$ was aminoacylated by incubation of 30 or 120 pmol of tRNA, respectively, with 50 mM Tris pH 8, 25 mM KCl, 100 mM NH$_4$Cl, 10 mM MgCl$_2$, 1 mM DTT, 5 mM ATP, 0.5 mM EDTA, 5 mM Met amino acid, 600 nM Met-RS in a 10–20 μl reaction at 37 °C for 15 min.

Human 80 S initiation complexes for smFRET and cryo-EM were formed by incubating 20 or 100 pmol of small subunits (heat-activated at 40 °C for 5 min), respectively, and 4×molar excess of prepared mRNA/DNA duplex (smFRET) or mRNA (cryo-EM) in 80 S polymix for 10 min at 37 °C and for 5 min on ice. The freshly prepared aminoacylated tRNA was added, followed by incubation for 10 min at 37 °C and for 5 min on ice. Equimolar large subunits (heat-activated at 40 °C for 5 min) were added (final reaction volume 50–100 μl) and the reaction mixture was incubated for 20 min at 37 °C and for 5 min on ice. MgCl$_2$ concentration was then adjusted to 15 mM and the reaction mixture was loaded onto a 10–30% sucrose gradient prepared in 80 S polymix as above but with 15 mM MgCl$_2$. Gradients were centrifuged at 150 krcf, 4 °C for 90 min and analyzed as described above, collecting the fraction corresponding to 80 S complexes. For smFRET, the resulting fractions aliquoted and flash-frozen in liquid nitrogen. For cryo-EM analysis, 80 S initiation complexes were pelleted at 150 krcf, 4 °C for 90 min and resuspended in a minimal volume of 80 S polymix with 5 mM MgCl$_2$ for a final concentration of 2–3 μM.

## Purification of rabbit eEF1A

Rabbit reticulocyte lysate (Green Hectares) was thawed, supplemented with 1×Mammalian ProteaseArrest (G-Biosciences), 1 mM phenylmethane sulfonyl fluoride, and 500 mM KCl, layered onto a cushion of 20 mM Tris HCl pH 7.5, 1 M sucrose, 500 mM KCl, 5 mM MgCl$_2$, 1 mM DTT, 20% glycerol, and centrifuged for 14 hr at 125 krcf, 4 °C. The supernatant was fractionated by precipitation with increasing concentrations of (NH$_4$)$_2$SO$_4$ by gradual addition of saturated solution while stirring at 4 °C. The fraction corresponding to 30–40% saturation was centrifuged for 20 min at 25 krcf, 4 °C. Pellets were resuspended into and dialyzed against Buffer A (20 mM Tris-HCl pH 7.5, 50 mM KCl, 0.1 mM EDTA, 0.25 mM DTT, 20% glycerol).

The sample was then further purified by three ion-exchange steps. In each step, fractions enriched in eEF1A were detected by Western Blot analysis (primary antibody: Millipore 05–235, used at 1:2000 dilution in TBST with 5% w/v dry milk). The columns used for the three steps were (i) DEAE FF HiPrep 16/10, (ii) SP HP 5 mL HiTrap, (iii) Mono S 5/50 GL (all from GE). In all three steps, Buffer A was used for column equilibration and sample loading; a gradient into Buffer B (identical to Buffer A but with 1 M KCl) was used for elution. In step (i), eEF1A was enriched in the flow-through, in subsequent steps, it was enriched in the eluted fractions. After steps (i) and (ii), the most enriched fraction as identified by Western Blot was dialyzed against Buffer A; after step (iii), the final product (~85% pure) was dialyzed against storage buffer (20 mM Tris HCl pH 7.5, 25 mM KCl, 6 mM BME, 5 mM Mg(OAc)$_2$, 60% glycerol) and stored at –20 °C.

## smFRET analysis of aa-tRNA selection

smFRET experiments were performed on a custom-built prism-type TIRF microscope as described previously (*Juette et al., 2016*). Briefly, biotinylated 80 S initiation complexes were immobilized in flow cells treated with a mixture of polyethylene glycol (PEG) and PEG-biotin and functionalized with streptavidin (*Blanchard et al., 2004b*). For aa-tRNA selection experiments, ternary complex containing *E. coli* Phe-tRNA$^{Phe}$ labeled with Cy5 at position acp$^3$U47 (*Blanchard et al., 2004b*), rabbit eEF1A, and GTP was delivered by manual injection at a final concentration of 20 nM. All experiments were performed in 80 S polymix buffer with 5 mM MgCl$_2$ as above. A 532 nm diode-pumped solid-state laser (Opus, LaserQuantum) was used for Cy3 excitation, fluorescence was collected through a 60×/1.27 NA water-immersion objective (Nikon), spectrally separated using a T635lpxr-UF2 dichroic mirror (Chroma) and imaged onto two cameras (Orca-Flash 4.0 v2, Hamamatsu). Time resolution was 15ms for all experiments except for the sneak-through/wash-out experiments shown in *Figure 2F–I*, which were performed at 1 s time resolution.

## smFRET data processing and analysis

Single-molecule fluorescence and FRET traces were extracted and further analyzed using our freely available MATLAB-based software platform SPARTAN (*Juette et al., 2016*) (http://scottcblanchardlab.com/software), extended with custom scripts. For display of example traces (*Figure 1*) and all quantitative analysis, traces were idealized using the sequential k-means algorithm based on a hidden Markov model (*Qin, 2004*). For display of non-equilibrium aa-tRNA selection data, all detected events were post-synchronized by aligning them to the first appearance of FRET. FRET contour plots were generated by compiling two-dimensional histograms of FRET occupancy over time for all traces. For dose-response curves (*Figure 1E and F* and *Figure 1—figure supplement 1*), accommodated molecules were defined as events spending 300ms or more in high-FRET. EC$_{50}$ values were obtained by fitting a Hill equation to the accommodated fraction as a function of drug concentration. To assess kinetic differences during aa-tRNA selection (*Figure 2*), the analysis of each idealized FRET trace was restricted to the time interval prior to the first dwell in high FRET lasting 150ms or more (shorter than the above-mentioned definition of accommodated molecules to reduce contributions from molecules achieving hybrid states after accommodation). These truncated, idealized traces were used for the computation of state lifetimes and transition ratios. Error bars in *Figure 1E and F*, *Figure 1—figure supplement 1*, and *Figure 2* represent standard errors obtained by bootstrap analysis (1000 samples) of the pooled data from all experimental repeats.

## Rabbit in vitro translation and cryo-EM sample preparation

In vitro translation reactions of a transcript encoding 3×Flag-tagged KRas were performed in a rabbit reticulocyte lysate (RRL) system at 32 °C as previously described (*Shao et al., 2016*; *Sharma et al., 2010*). A final concentration of 50 µM ternatin-4 was added after 7 min to stall ribosome-nascent chain complexes (RNCs) at the stage of aa-tRNA delivery by eEF1A and the reaction allowed to proceed to 25 min. A 4 ml translation reaction was directly incubated with 100 µl (packed volume) of anti-Flag M2 beads (Sigma) for 1 hr at 4 °C with gentle mixing. The beads were washed sequentially with 6 ml of buffer (50 mM HEPES pH 7.4, 5 mM Mg(OAc)$_2$, and 1 mM DTT) containing the additional components as follows: (1) 100 mM KOAc and 0.1% Triton X-100; (2) 250 mM KOAc 0.5% and Triton X-100; (3, RNC buffer) 100 mM KOAc. Two sequential elutions were carried out with 100 µl 0.1 mg/ml 3×Flag peptide (Sigma) in RNC buffer at room temperature for 25 min. The elutions were combined and centrifuged

at 100,000 rpm at 4 °C for 40 min in a TLA120.2 rotor (Beckman Coulter) before resuspension of the ribosomal pellet in RNC buffer containing 1 µM ternatin-4. The resuspended RNCs were adjusted to 120 nM and directly frozen to grids for cryo-EM analysis.

R2/2 grids (Quantifoil) were covered with a thin layer of continuous carbon (estimated to be 50 Å thick) and glow discharged to increase hydrophilicity. The grids were transferred to a Vitrobot MKIII (FEI) with the chamber set at 4 °C and 100% ambient humidity. Aliquots of purified RNCs (3 µl, ~120 nM concentration in 50 mM HEPES pH 7.4, 100 mM KOAc, 5 mM Mg(OAc)$_2$, 1 mM DTT and 1 µM ternatin-4) were applied to the grid and incubated for 30 s, before blotting for 3 s to remove excess solution, and vitrified in liquid ethane.

## Sample preparation for human cryo-EM structure determination

Gold R1.2/1.3 300 mesh grids (UltrAuFoil) were plasma cleaned (ArO$_2$, 7 s) and transferred to a Vitrobot MKII (FEI) with the chamber set at 4 °C and 100% ambient humidity. Aliquots of purified human 80 S initiation complexes in 80 S polymix buffer were thawed and brought to 0.2 µM didemnin B (didemnin) or 20 µM ternatin-4. Ternary complex containing *E. coli* Phe-tRNA$^{Phe}$, rabbit eEF1A, GTP, and either 0.2 µM didemnin or 20 µM ternatin-4 were added to 80 S initiation complexes for final concentrations of ~200 nM 80 S and aa-tRNA. aa-tRNA selection reactions were applied to the grid (3 µl) and incubated for ~45 s, before blotting for 2–3 s to remove excess solution, and vitrified in liquid ethane.

## Cryo-EM data collection and image processing for the rabbit structure

All micrographs of rabbit 80 S ribosomes were taken on an FEI Titan Krios microscope (300 kV) equipped with an FEI Falcon II direct-electron detector using quasi-automated data collection (EPU software, FEI). Movies were recorded at a magnification of ~135,000 ×, which corresponds to the calibrated pixel size of 1.04 Å per pixel at the specimen level. During the 1 s exposure, 40 frames (0.06 s per frame) were collected with a total dose of around 40 e⁻ per Å$^2$. Movie frames were aligned using whole-image motion correction (*Li et al., 2013*). Parameters of the contrast transfer function (CTF) for each motion-corrected micrograph were obtained using Gctf (*Zhang, 2016*). Visual inspection of the micrographs and their corresponding Fourier transforms was used to remove micrographs due to astigmatism, charging, contamination, and/or poor contrast.

Ribosome particles were selected from the remaining micrographs using semi-automated particle picking implemented in RELION 1.4 (*Scheres, 2015*). Reference-free two-dimensional class averaging was used to discard non-ribosomal particles. The retained particles underwent an initial three-dimensional refinement using a 30 Å low-pass filtered cryo-EM reconstruction of the didemnin-stalled rabbit ribosomal elongation complex (EMDB-4130) as an initial model. After refinement, the particles were then subjected to three-dimensional classification to separate different compositions and conformations of the ribosome complexes and isolate particles with high occupancy of the desired factors. From this classification, particles containing P- and E-site tRNAs were selected and re-refined. The movement of each particle within this subset was further corrected using RELION 1.4 (*Scheres, 2015*). The resulting 'shiny' particles were subjected to focused classification with signal subtraction (FCwSS) (*Bai et al., 2015*) to isolate particles containing pre-accommodated aa-tRNA and eEF1A. An additional round of 3D refinement was used to obtain the final map, which reached an overall resolution of 4.1 Å based on the Fourier shell correlation (FSC) 0.143 criterion (*Rosenthal and Henderson, 2003*). During post-processing, high-resolution noise substitution was used to correct for the effects of a soft mask on FSC curves (*Chen et al., 2013*) and density maps were corrected for the modulation transfer function (MTF) of the Falcon II detector and sharpened by applying a negative B-factor that was estimated using automated procedures (*Rosenthal and Henderson, 2003*). See *Figure 3—figure supplement 1* and *Table 2* for details.

## Cryo-EM data collection and image processing for the human structures

All micrographs of human 80 S ribosomes were taken on an FEI Titan Krios microscope (300 kV) equipped with an Gatan K2 Summit direct electron detector using Leginon MSI (*Suloway et al., 2005*) data collection. For didemnin, movies were recorded in counting mode at a magnification of 105,000 × (~1.072 Å$^2$ per pixel) with 10 s exposure for 50 frames (0.2 s per frame) with a total dose of around 67 e⁻ per Å$^2$. For ternatin-4, movies were recorded in super resolution mode at a magnification of

105,000 × (~1.096 Å² per pixel after 2×binning) with 10 s exposure for 50 frames (0.2 s per frame) with a total dose of around 70 e⁻ per Å². See *Table 2* for details. Movie frames were aligned using whole-image motion correction (*Li et al., 2013*). CTF parameters for each motion-corrected micrograph were obtained using CTFFIND4 (*Rohou and Grigorieff, 2015*). Visual inspection of the micrographs and their corresponding Fourier transforms was used to remove micrographs due to astigmatism, charging, contamination, and/or poor contrast.

Ribosome particles were selected from the remaining micrographs using semi-automated particle picking implemented in RELION 2.0 (*Kimanius et al., 2016*). Reference-free two-dimensional class averaging was used to discard non-ribosomal particles. The remaining particles were subjected to two rounds of three-dimensional classification with alignment to separate different compositions and conformations of the ribosome complexes, sorting first for 80 S particles followed by sorting for rotated and unrotated small ribosomal subunits. Particles with unrotated subunits were subjected to focused classification and FCwSS (*Bai et al., 2015*) to isolate particles containing pre-accommodated aa-tRNA and eEF1A. An additional round of 3D refinement was used to obtain the final maps, which reached overall resolutions of 3.2 Å and 3.8 Å for didemnin and ternatin-4, respectively, based on the FSC 0.143 criterion (*Rosenthal and Henderson, 2003*). During post-processing, noise substitution was used to correct for the effects of a mask on FSC curves (*Chen et al., 2013*) and density maps were corrected for the MTF of the K2 detector and sharpened by applying a –20 B-factor and a 4 Å low pass filter (*Rosenthal and Henderson, 2003*). See *Figure 3—figure supplement 3* and *Table 2* for details.

## Cryo-EM map interpretation

Density map values were normalized to mean = 0 and standard deviation (σ)=1 in UCSF Chimera (*Pettersen et al., 2004*) using the vop scale function. The pixel size of each map was calibrated against a 2.9 Å resolution structure of the mammalian ribosome (PDB-ID: 6QZP) (*Natchiar et al., 2017*) and maps were aligned on the large subunit core. The model of the rabbit ribosomal elongation complex trapped with didemnin (PDB-ID: 5LZS) (*Shao et al., 2016*) was docked into the cryo-EM maps with Chimera (*Pettersen et al., 2004*). Density present in the didemnin binding site was interpreted as belonging to didemnin or ternatin-4, respectively. A model for ternatin-4 was rigid-body fit into the cryo-EM density for figure images. However, as the density was insufficiently resolved to unambiguously place a model for ternatin-4, the density was left unmodelled.

## Cell lines

HCT116 cells (ATCC) were maintained in McCoy's 5 A media (Gibco) supplemented with 10% FBS (Axenia Biologix), 100 units ml⁻¹ penicillin and 100 µg ml⁻¹ streptomycin (Gibco). Jurkat cells (ATCC) were maintained in RPMI 1640 media (Gibco) supplemented with 10% FBS (Axenia Biologix), 100 units ml⁻¹ penicillin and 100 µg ml⁻¹ streptomycin. All cells were cultured at 37 °C in a 5% CO₂ atmosphere. Cell lines were tested for mycoplasma (Lonza) before use and were SNP-typed to verify that their identity matched that of the originally procured sample. None of the tested lines are present in the list of cross-contaminated/misidentified cell lines.

## Homoprogargyl glycine metabolic labeling

HCT116 cells were seeded in 24-well plates at 30,000 cells/well and incubated overnight before 4 hr treatment with compound. For experiments involving washout, cells were washed twice with 1 ml complete media, followed by alternating quick and 5 min 37 °C washouts repeated four times each (*O'Hare et al., 2013*). After appropriate incubations, cells were washed once with phosphate-buffered saline (PBS), then exchanged to methionine- and cysteine-free DMEM (Gibco) supplemented with 10% dialyzed FBS (Axenia Biologix), glutamine (2 mM), cysteine (2 mM), homopropargyl glycine (1 mM; Kerafast), and appropriate drug for 1 hr. Media was then aspirated, cells were trypsinized and transferred to 96-well plates, washed once with PBS, and fixable live/dead stained with Zombie Red amine-reactive dye (BioLegend) according to the manufacturer's instructions. Cells were fixed in 2% paraformaldehyde in PBS for 10 min at room temperature, and then permeabilized in PBS supplemented with 0.1% saponin and 3% FBS. Samples in 25 µl permeabilization buffer were subjected to copper-catalyzed alkyne-azide conjugation to CF405M-azide (Biotium) by addition of 100 µl click reaction mix (50 mM HEPES pH 7.5, 150 mM NaCl, 400 µM TCEP, 250 µM TBTA, 200 µM CuSO₄, 5 µM azide). After overnight incubation at room temperature in the dark, samples were washed 3×with

permeabilization buffer, 2×FACS buffer (PBS –Mg/Ca +2% FBS+2 mM EDTA) and analyzed by flow cytometry (MACSQuant VYB). Data analysis was performed with FlowJo software (Tree Star). Dead cells (Zombie Red +) were excluded from analysis (typically representing <15% of total cells), leaving at least 500 live cells for each data point, but additional cells were analyzed when possible (up to 10,000). $IC_{50}$ values and plotted data points (*Figure 5*) are given as the mean of three independent determinations ± standard error.

## Apoptosis assay by annexin V/propidium iodide staining

Jurkat cells at $0.5×10^6$ cells/ml were treated with compound as indicated. For experiments involving washout, cells were washed twice with 1 ml complete media by pelleting cells for 3 min at $1.3k×g$ and aspirating media, followed by alternating quick and 5 min 37 °C washouts repeated four times each (*O'Hare et al., 2013*). Cells were stained with annexin V-FITC and propidium iodide (BD Pharmingen) according to the manufacturer's instructions and analyzed by flow cytometry (MACSQuant VYB). The mean of three independent experiments ± standard error is plotted. For each drug treatment condition, 10,000 cells were analyzed.

## Flag-eEF1A purification

HCT116 cells stably expressing Flag-eEF1A1(WT or A399V)-P2A-mCherry (*Carelli et al., 2015*) were lysed in buffer containing 50 mM HEPES pH 7.5, 125 mM KOAc, 5 mM $MgOAC_2$, 1% Triton X-100, 10% glycerol, 1 mM DTT, and 1×EDTA free complete protease inhibitors (Roche). Lysate (8 mg/sample) was incubated with 200 μl anti-Flag magnetic beads (Sigma) at 4 °C for 90 min. Beads were washed three times with lysis buffer, three times with lysis buffer +400 mM KOAc, and three times with elution buffer (50 mM HEPES pH 7.5, 0.1 mM EDTA, 100 mM KCl, 25% glycerol, 1 mM DTT), then eluted in 100 mM elution buffer +1 mg/ml 3×Flag peptide (Sigma) at 4 °C for 30 min.

## Figure preparation

All figures containing cryo-EM density were generated with UCSF Chimera (*Pettersen et al., 2004*) or UCSF ChimeraX (*Pettersen et al., 2021*). Density was colored using the Color Zone tool in UCSF ChimeraX with a 3 Å radius. Maps colored by local resolution were visualized in Chimera using LocalRes in RELION 4.0 (*Kimanius et al., 2021*; *Figure 3—figure supplements 1 and 3*). All figures were compiled in Adobe Illustrator (Adobe).

## Rigor and reproducibility

Biological replicates are defined here as independent measurements of physically distinct samples. Technical replicates are defined as repeated measurements of the same physical sample. eEF1A used for smFRET and cryo-EM experiments was purified from two distinct batches of RRL. For smFRET experiments, the number of traces (N) is indicated in each figure panel and all experiments were performed in biological triplicate and were repeated on different days. Error bars in smFRET experiments represent standard errors obtained by bootstrap analysis (1000 samples) of the pooled data from all experimental repeats. The sample size chosen for bootstrap analysis converged to within the obtainable precision for the experimental setup, that is larger samples would have incurred more computation time without yielding additional information. For homoprogargyl glycine metabolic labeling assays and apoptosis assays, $IC_{50}$ values and plotted data points are given as the mean of three independent biological determinations ± standard error.

## Acknowledgements

This work was funded by the Howard Hughes Medical Institute (JT) and by National Institutes of Health (NIH) grants GM079238 to SCB and GM115327-Tan to EJR. We thank D Terry, R Altman, and other members of the Blanchard laboratory for their expertise and efforts to enable the single-molecule investigations performed and for their review of the manuscript during the preparation and completion of this research. We acknowledge support from the Single-Molecule Center at St Jude Children's Research Hospital. We thank A Plante at Weill Cornell Medicine for guidance in data processing and high-performance computing. Some of this work was performed at the Simons Electron Microscopy Center (SEMC) located at the New York Structural Biology Center, supported by grants from the Simons Foundation (349247), NYSTAR, and the NIH (GM103310). We acknowledge

specific support from SEMC members B Carragher, C Potter, E Eng, and D Bobe for guidance with cryo-EM grid preparation and data collection.

## Additional information

### Competing interests
Jack Taunton: JT is listed as an inventor on a patent application covering ternatin analogs (PCT/US2021/016790, patent pending). Scott C Blanchard: SCB holds an equity interest in Lumidyne Technologies. The other authors declare that no competing interests exist.

### Funding

| Funder | Grant reference number | Author |
|---|---|---|
| National Institutes of Health | GM079238 | Scott C Blanchard |
| National Institutes of Health | GM115327-Tan | Emily J Rundlet |
| Howard Hughes Medical Institute | | Jack Taunton |

The funders had no role in study design, data collection and interpretation, or the decision to submit the work for publication.

### Author contributions
Manuel F Juette, Data curation, Software, Formal analysis, Investigation, Methodology, Writing – original draft, Writing – review and editing, Performed and analyzed smFRET experiments; Jordan D Carelli, Data curation, Formal analysis, Investigation, Methodology, Writing – original draft, Writing – review and editing, Performed cellular experiments; Emily J Rundlet, Data curation, Formal analysis, Validation, Investigation, Visualization, Methodology, Writing – original draft, Writing – review and editing, Performed cryo-EM studies on the human 80S ribosome and created figure panels and illustrations; Alan Brown, Data curation, Formal analysis, Investigation, Visualization, Methodology, Writing – review and editing, Performed cryo-EM studies on the rabbit 80S ribosome; Sichen Shao, Data curation, Formal analysis, Investigation, Methodology, Writing – review and editing, Performed cryo-EM studies on the rabbit 80S ribosome; Angelica Ferguson, Formal analysis, Investigation, Methodology, Performed cryo-EM studies on the human 80S ribosome; Michael R Wasserman, Formal analysis, Investigation, Performed and analyzed smFRET experiments; Mikael Holm, Formal analysis, Validation, Writing – review and editing, Performed and analyzed smFRET experiments; Jack Taunton, Scott C Blanchard, Conceptualization, Data curation, Supervision, Funding acquisition, Writing – original draft, Writing – review and editing

### Author ORCIDs

Manuel F Juette http://orcid.org/0000-0002-8760-8080
Jordan D Carelli http://orcid.org/0000-0001-7625-7505
Emily J Rundlet http://orcid.org/0000-0003-4041-6206
Alan Brown http://orcid.org/0000-0002-0021-0476
Sichen Shao http://orcid.org/0000-0003-2679-5537
Scott C Blanchard http://orcid.org/0000-0003-2717-9365

### Decision letter and Author response
Decision letter https://doi.org/10.7554/eLife.81608.sa1
Author response https://doi.org/10.7554/eLife.81608.sa2

## Additional files

### Supplementary files
• MDAR checklist

• Transparent reporting form

## Data availability

MATLAB-based software platform for smFRET analysis SPARTAN (Juette et al., 2016) is freely available at http://scottcblanchardlab.com/software. Cryo-EM 3D maps for all structures are available through the Electron Microscopy Data Bank (EMDB) as follows: ternatin-4-stalled elongating rabbit 80S, EMD-27732; didemnin B-stalled human 80S initiation complex, EMD-27691; ternatin-4-stalled human 80S initiation complex, EMD-27694.

The following datasets were generated:

| Author(s) | Year | Dataset title | Dataset URL | Database and Identifier |
|---|---|---|---|---|
| Rundlet EJ, Ferguson A, Juette MF, Carelli JD, Taunton J, Blanchard SC | 2022 | eEF1A(GDP)aa-tRNA stalled on the human 80S ribosome by ternatin-4 | https://www.ebi.ac.uk/embdb/EMD-27694 | EMDB, EMD-27694 |
| Rundlet EJ, Ferguson A, Juette MF, Carelli JD, Taunton J, Blanchard SC | 2022 | eEF1A(GDP)aa-tRNA stalled on the human 80S ribosome by didemnin B | https://www.ebi.ac.uk/embdb/EMD-27691 | EMDB, EMD-27691 |
| Brown A, Shao S, Rundlet EJ, Juette MF, Carelli JD, Taunton J, Blanchard SC | 2022 | eEF1A(GDP)aa-tRNA stalled on the rabbit 80S ribosome by ternatin-4 | https://www.ebi.ac.uk/embdb/EMD-27732 | EMDB, EMD-27732 |

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
