## [Editor Report]

Juette and coworkers employed single-molecule fluorescence, cryogenic-electron microscopy structures, and in vivo measurements to investigate the mechanism whereby two natural products with potential as cancer therapeutics (didemnin B and ternatin-4) inhibit accommodation of tRNA within the ribosomal A site during translation elongation. Their results demonstrate convincingly that both molecules inhibit tRNA accommodation by interfering with the movement of eukaryotic elongation factor 1 α after its activation by the GTPase activation site of the ribosome, but that the degree and nature of this restriction are subtly different between the two, leading to more marked differences in their effects on global translation and cell growth. The results of this important and valuable interdisciplinary work solidify prior conclusions, particularly on didemnin B, and illuminate the similarities and differences in how these two drugs interfere with the normal functioning of the elongating ribosome in vitro and inhibit protein synthesis and cell growth in vivo.

---

## [Decision Letter]

**Decision letter after peer review:**

Thank you for submitting your article "Didemnin B and ternatin-4 inhibit conformational changes in eEF1A required for aminoacyl-tRNA accommodation into mammalian ribosomes" for consideration by *eLife*. Your article has been reviewed by 3 peer reviewers, and the evaluation has been overseen by a Reviewing Editor and James Manley as the Senior Editor. The following individuals involved in review of your submission have agreed to reveal their identity: Colin Echeverría Aitken (Reviewer #1); Joseph D Puglisi (Reviewer #2); Sarah E Walker (Reviewer #3).

Essential revisions:

– An improved presentation of the structural data is required, including revamping Figure 4 and possibly inserting a new figure to present more details on drug interactions with eEF1A. It's not clear from the structural data why distinct mechanisms are observed for the two drugs, and the authors should attempt to present a more detailed description and analysis of the drug interactions with eEF1A that might address this point.

– Revise the text to provide more explanation In Results of the questions being asked, the experimental approach being used to answer them, and a consideration of different possible outcomes; coupled with deferring detailed discussions of results to the Discussion.

– The presentation of Figure 2 would benefit from adding schematics of the different reaction states, and a more transparent summary of the data is also required.

– Non-specialists would benefit from introductory explanations of classical and hybrid conformational states of the ribosome.

– Add text to the Discussion concerning the implications of the work for targeting translation elongation in human disease.

– Minimize the use of abbreviations to increase readability.

– Respond with the appropriate revisions to all other points raised in the reviewers' recommendations.

*Reviewer #1 (Recommendations for the authors):*

I believe this to be a rigorous and well reasoned study addressing important questions in our understanding of translation mechanism and how these are affected by small molecule inhibitors. Generally, the interpretation of experimental data is sound and reasonable. That said, much of this interpretation leaks into the Results section (for examples lines 232-240 or lines 319-323 … but there are others), whereas the Discussion section is rather abrupt. I would suggest moving this interpretation to the Discussion section, where it avoids influencing the reader's own interpretation of the data. To make these data accessible, the authors might consider replacing this interpretation with a more clear presentation of the experimental approach being taken, how it addresses the particular question being asked, and what possible results might be expected under different potential scenarios. In addition, a more complete and accessible presentation of prior results related to each experiment, whether during the Results section or later in the discussion, might make the relationship between these results and the experiments being presented (or the conclusions being drawn) more accessible to the reader (for example during the discussion of the CHX effects on hybrid/classical dynamics and how this enables on/off pathway determination). Overall, I believe this manuscript to be worthy of publication with revisions, given its rigorous treatment of important questions in the translation field, and its presentation of telling data from multiple techniques that shed new light on those questions.

*Reviewer #2 (Recommendations for the authors):*

Overall, this is a rigorous and well performed study probing the mechanisms of drug action in human translation elongation. The combination of dynamics measurements and structure are particularly novel, and will complement ongoing investigations (and publications) by the Blanchard lab on human elongation in general. As such, despite prior structural work on these drugs, the manuscript deserves publication in *eLife*, upon addressing some concerns outlined below.

1. The biggest issue is the structural data and Figure 4 in the manuscript. It is really hard to see the dynamic differences in density in the panels. To me the main point of this figure should be the overall drug binding site, molecular details of the drug interaction, lack of π density and the differences in EM density between the two drug complexes (which are at different overall resolutions). As this figure is key to the conclusions of the manuscript, I would recommend "starting from square zero" to clarify and simplify this figure. For example, I liked supp figure 3 which highlights the interactions and global maps.

2. In that regard, I would love to see details of the drug interaction with eEF1A, and understand the SAR of whey ternatin-4 has these different properties. Are there hints from this work on how to make this class of compounds better?

3. Finally, as this is a fine piece of work on inhibitors of human elongation, how do the authors view targeting of this process in human disease. Is this just like traditional chemotherapy that targets increased protein synthesis demands or is there another path forward for specificity?

4. For Figure 2, it would be nice to have a reaction schematic of the states and then a mini-table of the key rates from the single-molecule experiments that are buried in Table S1.

---

## [Author Response]

Essential revisions:– An improved presentation of the structural data is required, including revamping Figure 4 and possibly inserting a new figure to present more details on drug interactions with eEF1A. It's not clear from the structural data why distinct mechanisms are observed for the two drugs, and the authors should attempt to present a more detailed description and analysis of the drug interactions with eEF1A that might address this point.

We have made substantial updates to Figures 3 and 4 to improve the presentation of the cryo-EM data. This includes a more detailed description of the drug binding pocket (Figure 3), and a more focused description of the differences we observed in the G domain when eEF1A is stalled with didemnin or ternatin-4. The local resolution of ternatin-4 prevented us from unambiguously modeling its orientation and contact points in the drug binding pocket, but we have, nonetheless, presented hypotheses linking differential interactions with eEF1A to differential activity in vitro and in vivo as compared to didemnin. These can be found in the results and Discussion sections.

– Revise the text to provide more explanation In Results of the questions being asked, the experimental approach being used to answer them, and a consideration of different possible outcomes; coupled with deferring detailed discussions of results to the Discussion.

We have updated each Results section in the manuscript to more clearly define the questions and objectives of each experiment. We have also sequestered discussion points to the Discussion section.

– The presentation of Figure 2 would benefit from adding schematics of the different reaction states, and a more transparent summary of the data is also required.

We have added a schematic and a small summary of Table S1 to Figure 2 and have updated the text to add clarity and context for the reader.

– Non-specialists would benefit from introductory explanations of classical and hybrid conformational states of the ribosome.

This has been added to the text and to schematics in Figures 1 and 2.

– Add text to the Discussion concerning the implications of the work for targeting translation elongation in human disease.

We have expanded the discussion to include these points.

– Minimize the use of abbreviations to increase readability.

We have removed 6 abbreviations from the manuscript.

– Respond with the appropriate revisions to all other points raised in the reviewers' recommendations.

Please find below a point-by-point reply to the requested revisions below.

Reviewer #1 (Recommendations for the authors):I believe this to be a rigorous and well reasoned study addressing important questions in our understanding of translation mechanism and how these are affected by small molecule inhibitors. Generally, the interpretation of experimental data is sound and reasonable. That said, much of this interpretation leaks into the Results section (for examples lines 232-240 or lines 319-323 … but there are others), whereas the Discussion section is rather abrupt. I would suggest moving this interpretation to the Discussion section, where it avoids influencing the reader's own interpretation of the data.

We have worked to sequester interpretation of results to the Discussion section, while maintaining summarizing statements to help the reader understand the findings.

To make these data accessible, the authors might consider replacing this interpretation with a more clear presentation of the experimental approach being taken, how it addresses the particular question being asked, and what possible results might be expected under different potential scenarios.

We thank Reviewer 1 for outlining ways to make our approach and interpretations more accessible. We have added text to the introduction and to the beginning of each Results section to guide the reader through the questions being asked and how each experiment addresses said questions.

In addition, a more complete and accessible presentation of prior results related to each experiment, whether during the Results section or later in the discussion, might make the relationship between these results and the experiments being presented (or the conclusions being drawn) more accessible to the reader (for example during the discussion of the CHX effects on hybrid/classical dynamics and how this enables on/off pathway determination).

We have expanded our explanations of terminology and processes that, while known within the field, will likely be new to the general reader. This can be found in the introduction, the smFRET results section, and the discussion. We have also added a schematic to Fig. 2 to better illustrate our results and interpretations.

Reviewer #2 (Recommendations for the authors):Overall, this is a rigorous and well performed study probing the mechanisms of drug action in human translation elongation. The combination of dynamics measurements and structure are particularly novel, and will complement ongoing investigations (and publications) by the Blanchard lab on human elongation in general. As such, despite prior structural work on these drugs, the manuscript deserves publication in eLife, upon addressing some concerns outlined below.1. The biggest issue is the structural data and Figure 4 in the manuscript. It is really hard to see the dynamic differences in density in the panels. To me the main point of this figure should be the overall drug binding site, molecular details of the drug interaction, lack of π density and the differences in EM density between the two drug complexes (which are at different overall resolutions). As this figure is key to the conclusions of the manuscript, I would recommend "starting from square zero" to clarify and simplify this figure. For example, I liked supp figure 3 which highlights the interactions and global maps.

We agree with Reviewer 2 that the cryo-EM density in Figures 3 and 4 was difficult to see in mesh representation. We also agree that our discussion points in the text were not illustrated as well as they could be in the figures, thus we have made substantial updates to Figures 3 and 4 to address these critiques.

We have added new panels to Figure 3 to more concisely illustrate the interactions between didemnin/ternatin-4 and eEF1A (see point (2) below). For Figure 4, which focuses on the G-domain, we added an inset to show density for GDP in the nucleotide binding pocket of the ternatin-4 stalled structure. We have changed all density to surface representation to more clearly illustrate our points (lack of density in the ternatin-4-stalled vs. the didemnin-stalled structure). We have also removed cartoon in regions with poor density and modified the camera angles to show the change in switch I contact points in the didemnin vs ternatin-4 stalled structure. We hope Reviewer 2 will agree that these changes greatly improved the interpretability of our results.

2. In that regard, I would love to see details of the drug interaction with eEF1A, and understand the SAR of whey ternatin-4 has these different properties. Are there hints from this work on how to make this class of compounds better?

We, too, would love to know the specific details of the interactions between eEF1A and teratin-4! But, as the quality of the cryo-EM density in this region prohibited unambiguous assignment of the ternatin-4 orientation, we opted not to detail the specific contacts made between ternatin-4 and eEF1A to avoid overinterpretation of our data. We have, however identified regions of eEF1A that are contacted by didemnin but not by ternatin-4, which could help explain the observed differences in dynamics and in vivo experimentation. This could also be of value in derivatization efforts of ternatin-4 by medicinal chemists. We have added these observations and hypotheses to the results and Discussion sections, respectively.

3. Finally, as this is a fine piece of work on inhibitors of human elongation, how do the authors view targeting of this process in human disease. Is this just like traditional chemotherapy that targets increased protein synthesis demands or is there another path forward for specificity?

We thank Reviewer 2 for outlining an opportunity to comment on the utility of targeting translation for the treatment of human disease. We have now expanded this topic in the revised discussion.

4. For Figure 2, it would be nice to have a reaction schematic of the states and then a mini-table of the key rates from the single-molecule experiments that are buried in Table S1.

We thank Reviewer 2 for these suggestions and have added both requested elements to Figure 2.